# A combination of plasma membrane sterol biosynthesis and autophagy is required for shade-induced hypocotyl elongation

Yetkin Çaka Ince [1], Johanna Krahmer[1], Anne-Sophie Fiorucci [1], Martine Trevisan [1], Vinicius Costa Galvão[1], Leonore Wigger [2], Sylvain Pradervand[2], Laetitia Fouillen [3], Pierre Van Delft [3], Manon Genva[3,4], Sebastien Mongrand[3], Hector Gallart-Ayala[5], Julijana Ivanisevic [5] & Christian Fankhauser [1] ✉

Plant growth ultimately depends on fixed carbon, thus the available light for photosynthesis. Due to canopy light absorption properties, vegetative shade combines low blue (LB) light and a low red to far-red ratio (LRFR). In shade-avoiding plants, these two conditions independently trigger growth adaptations to enhance light access. However, how these conditions, differing in light quality and quantity, similarly promote hypocotyl growth remains unknown. Using RNA sequencing we show that these two features of shade trigger different transcriptional reprogramming. LB induces starvation responses, suggesting a switch to a catabolic state. Accordingly, LB promotes autophagy. In contrast, LRFR induced anabolism including expression of sterol biosynthesis genes in hypocotyls in a manner dependent on PHYTOCHROME-INTERACTING FACTORs (PIFs). Genetic analyses show that the combination of sterol biosynthesis and autophagy is essential for hypocotyl growth promotion in vegetative shade. We propose that vegetative shade enhances hypocotyl growth by combining autophagy-mediated recycling and promotion of specific lipid biosynthetic processes.

Plants use a portion of the electromagnetic spectrum for photosynthesis that is called photosynthetically active radiation (PAR, 400–700 nm) and composed of blue (B, 400–500 nm), green (G, 500–600 nm) and red (R, 600–700 nm) light[1]. Leaves absorb more than 90% of B and R radiation, whereas they transmit and/or reflect most of the far-red light (FR, 700–760 nm)[2]. Therefore, plants under vegetative shade receive light combining low B (LB) and a low R/FR ratio (LRFR)[3]. In shade-avoiding plants, LB and LRFR independently trigger a suite of similar adaptive responses, including the growth of stem-like structures including hypocotyls and petioles to enhance light access[4–7]. Plants in

dense communities also receive LRFR due to FR reflection from neighboring leaves before actual shading. This is perceived as a neighbor proximity/shade threat signal and triggers similar growth adaptation as vegetative shade prior to declining light resources[3,4]. Plant growth depends on fixed carbon[8], which depends on PAR, including B and R light[2,9]. However, how LB and LRFR with contrasting carbon resource availability promote similar growth adaptations remains unclear[10].

While molecular mechanisms underlying hypocotyl growth promotion are relatively well-understood in LRFR, they remain unclear in LB[5–7]. LRFR inactivates phytochrome B (phyB), leading to

[1]Center for Integrative Genomics, Faculty of Biology and Medicine, Génopode Building, University of Lausanne, CH-1015 Lausanne, Switzerland. [2]Genomic Technologies Facility, Faculty of Biology and Medicine, Génopode Building, University of Lausanne, CH-1015 Lausanne, Switzerland. [3]Univ. Bordeaux, CNRS, Laboratoire de Biogenèse Membranaire, UMR 5200, F-33140 Villenave d'Ornon, France. [4]Laboratory of Chemistry of Natural Molecules, Gembloux Agro-Bio Tech, University of Liège, Passage des Déportés 2, 5030 Gembloux, Belgium. [5]Metabolomics Platform, Faculty of Biology and Medicine, Rue du Bugnon 19, University of Lausanne, CH-1005 Lausanne, Switzerland. ✉e-mail: christian.fankhauser@unil.ch

de-repression of phytochrome-interacting factors (PIFs) transcription factors (TFs)[5,6,11,12]. PIF7, with substantial contributions of PIF4 and PIF5, enhances auxin production in cotyledons through induction of *YUC-CAs* (*YUC2*, *YUC5*, *YUC8*, and *YUC9*) coding for auxin biosynthesis enzymes[13–18]. Auxin is rapidly transported to the hypocotyl, where it locally induces elongation, presumably through a combined auxin and PIF transcriptional response[14,19–21]. Additional hormones, notably brassinosteroids (BR) are also important for LRFR-induced hypocotyl growth[14,22,23]. Indeed, the BR signaling factor BZR1, the auxin response factor ARF6 and PIF4 collectively regulate target gene expression[21]. PIFs are also important for LB-induced hypocotyl elongation, where PIF4 is the primary one with contributions of PIF5 and PIF7[24–26]. The LB response is controlled by cryptochromes (cry), but how they regulate PIFs remains unclear. Cry1 inhibits PIF4 transcriptional activity following BL-induced interaction[26,27]. In LB cry2 interacts with PIF4/PIF5, but the functional importance of this complex remains unclear[26,27]. Despite auxin and BR being indispensable for hypocotyl elongation in LB, this is not apparent from the transcriptional response, which contrasts with LRFR conditions[24–26,28,29]. Despite these differences, several growth-related pathways are transcriptionally activated in LB and LRFR[14,26,28]. Nevertheless, the lack of spatial resolution limits our current understanding of LB vs LRFR growth-promoting mechanisms and the role of PIFs in hypocotyls.

Hypocotyl growth occurs by cell elongation where plasma membrane (PM) extension is essential[30–32]. Although the PM lipid bilayer can transit between tighter or looser packing depending on several parameters, the PM is fairly rigid with limited expansion or contraction ability[33]. Furthermore, PM curvature is low, also limiting its extension potential[34]. In rapidly elongating plant cells (e.g., pollen tubes and root hair cells), the PM grows with the deposition of new lipid material through the fusion of Golgi-derived vesicles carrying new cell wall material[31,32]. We previously reported that LRFR increases carbon allocation to lipids in *B. rapa* hypocotyls[35]. Furthermore, LRFR induces sterol biosynthesis gene expression in hypocotyls[14]. Sterols compose up to 30% of PM lipids[33]. They influence PM permeability and fluidity[33,36]. Together with sphingolipids, sterols are enriched in PM lipid microdomains that serve as anchoring platforms for signaling and transporting proteins[37,38]. Their major structural and functional roles at the PM suggest an important function of sterols in shade-induced hypocotyl elongation.

Production of new material required for cell elongation depends on carbon availability[8,39]. LRFR does not decrease carbon fixation in *B. rapa* seedlings, as PAR remains unchanged[35]. However, reducing PAR either by decreasing B, G, or R light decreases carbon fixation[2,9]. Thus, carbon fixation is expected to decrease in LB, limiting the availability of newly fixed carbon to sustain hypocotyl growth promotion. Carbon starvation triggered by transferring plants into darkness for several days induces autophagy that recycles unessential cytoplasmic materials by vacuolar hydrolases[40–42]. The fact that LB and LRFR differ in PAR suggests the deployment of different mechanisms to enable cell elongation in these distinct conditions.

Analyzing light-regulated gene expression in dissected hypocotyls was informative to understand hypocotyl growth regulation during de-etiolation and in LRFR[14,43], but we lack equivalent data for LB. Thus, we performed organ-specific gene expression to compare and contrast hypocotyl growth promotion in LB vs LRFR. We show that in LRFR PIFs induce expression of many anabolic processes in the hypocotyl, including sterol biosynthesis. In contrast, LB induces expression of catabolic processes and promotes autophagy, which is important for hypocotyl growth enhancement.

## Results

### LB and LRFR induce distinct transcriptional changes in elongating hypocotyls

Consistent with previous studies[24,26], LB and LRFR treatments independently induced hypocotyl elongation in a PIF and YUC-dependent manner (Fig. 1a). We hypothesized a convergence of LB and LRFR transcriptome in hypocotyls where both light conditions trigger cell elongation. Thus, we analyzed transcriptomes from dissected cotyledons and hypocotyls of Col-0 (wild type−WT) in white light (WL), LB, and LRFR to characterize the organ-specific LB and LRFR responses. *pif457* and *yuc2589* seedlings were used to determine the role of PIFs and YUC-mediated auxin biosynthesis (Fig. 1b).

Principal component analysis (PCA) showed that for biological replicates of each genotype, organ, and condition clustered closely (Supplementary Fig. 1a). In WT hypocotyls, LRFR induced more transcriptome changes than LB; while in cotyledons it was the opposite (Fig. 1c, Supplementary Fig. 1b, and Supplementary Data 1). The number of common up- and downregulated genes were higher in hypocotyls than in cotyledons (Fig. 1c and Supplementary Fig. 1b), suggesting a convergence of LB and LRFR transcriptome in elongating hypocotyls. This was confirmed by GO term enrichment analyses for LB-specific, LRFR-specific and shared LB and LRFR-upregulated genes (Fig. 1d, Supplementary Fig. 1c, and Supplementary Data 2). We highlighted selected terms for each organ and light condition that we could easily relate to growth regulation (Fig. 1d, Supplementary Fig. 1c, and full lists in Supplementary Data 2). Genes upregulated by both treatments in hypocotyls were enriched in terms related to cellular elongation including "growth", "cell wall organization or biogenesis", "exocytosis", "endocytosis", "cytoskeleton organization", "lipid biosynthetic process", and "response to brassinosteroid" (Fig. 1d). Genes specifically upregulated by LB in both organs were enriched in GO terms related to starvation (e.g., "cellular response to sucrose starvation") and catabolic events "protein catabolic process", "cellular lipid catabolic process" and "autophagy" (Fig. 1d and Supplementary Fig. 1c). In contrast, LRFR-specific genes in hypocotyls were enriched in biosynthetic processes including "ribosome biogenesis", "peptide biosynthetic process", "sterol biosynthetic process", and "cell wall organization and biogenesis" (Fig. 1d). In line with previous reports[14], LRFR induced many hormone related responses in cotyledons and "response to auxin" in hypocotyls (Fig. 1d and Supplementary Fig. 1c). Altogether, our GO term enrichment analyses indicate that LB and LRFR transcriptome responses converge in elongating hypocotyls, on the induction of several growth-related mechanisms. However, we observed a striking difference between these treatments with LB upregulating numerous catabolic processes and LRFR inducing many anabolic processes.

### Most LRFR-induced genes in hypocotyls require both PIFs and YUCs

In accordance with the established role of PIFs and YUCs for LRFR responses[13–18,44], gene expression in *pif457* and *yuc2589* was largely unresponsive to LRFR (Supplementary Fig. 1a). PCA showed that hypocotyls of LRFR-treated *pif457* and *yuc2589* grouped with WL samples but *yuc2589* cotyledons grouped closer to WT LRFR samples (Supplementary Fig. 1a). We also evaluated the individual roles of PIFs and YUCs by analyzing the interactions between genotypes and light treatments. These comparisons revealed four groups among LRFR-upregulated genes: PIF and YUC dependent, only PIF dependent, only YUC dependent, and dependent on neither (Fig. 2a, Supplementary Fig. 2a, and Supplementary Data 3). In hypocotyls, most LRFR-induced genes required PIFs and YUCs, whereas the largest fraction depended on only PIFs in cotyledons (Fig. 2a and Supplementary Fig. 2a). The extent of LRFR regulation was reduced in *pif457* and *yuc2589* also in the PIF- and/or YUC-independent categories (Fig. 2b and Supplementary Fig. 2b), indicating that our classification underestimates the importance of PIFs and YUCs. We conclude that LRFR-regulated gene expression largely depends on PIF4, PIF5 and/or PIF7. Moreover, while in cotyledons the regulation of many genes depends on PIFs alone, in the hypocotyl their regulation depends on PIFs and YUCs.

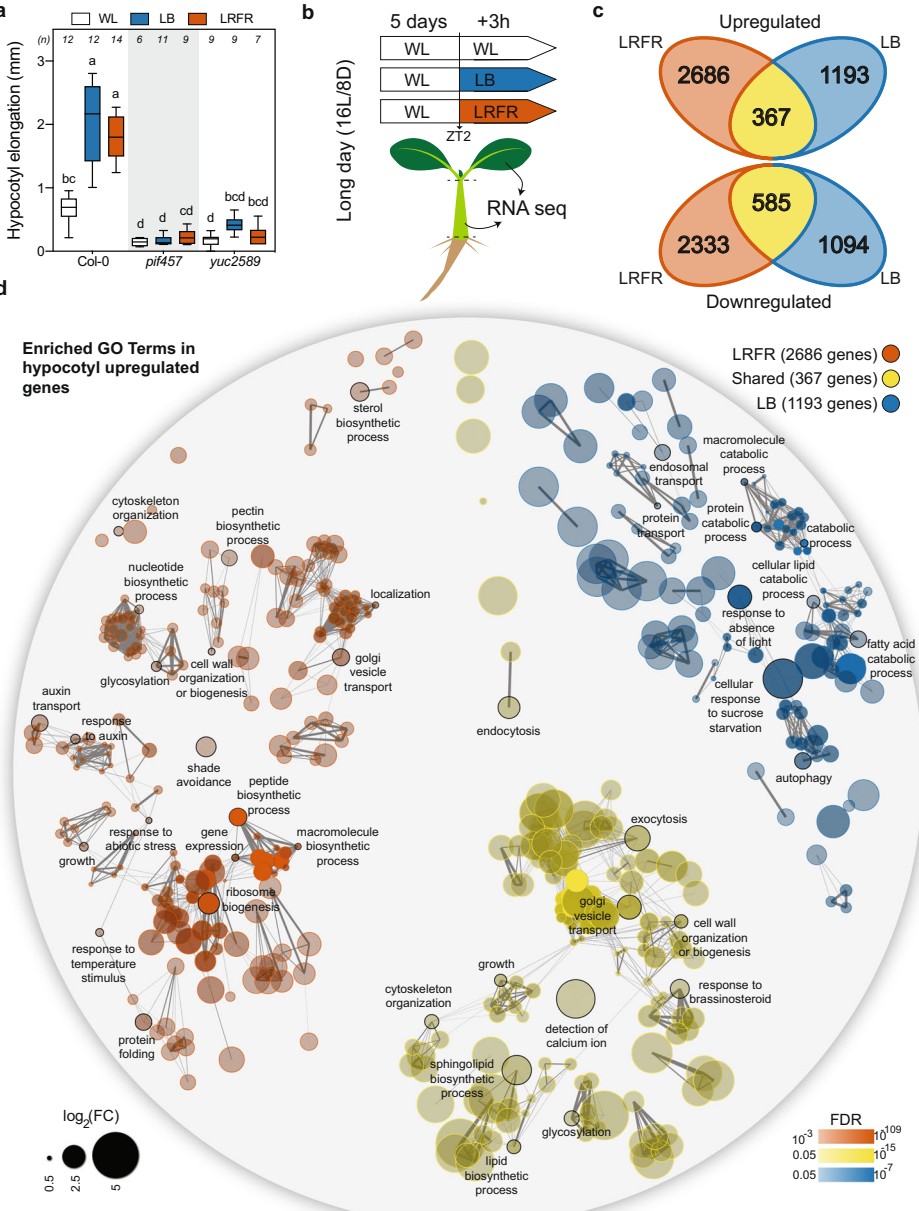

**Fig. 1 | LB and LRFR induce distinct transcriptional changes in elongating hypocotyls. a** Hypocotyl elongation of the indicated genotypes. The horizontal bar represents the median; boxes extend from the 25th to the 75th percentile, whiskers extend to show the data range. Different letters indicate significant differences ($P < 0.05$, two-way ANOVA with Tukey's HSD test; the exact $P$ values are available in the Source Data). Sample size ($n$) that is given on top indicates biologically independent seedlings examined over one experiment. The experiment was repeated two times with similar results. **b** Schematic summary of the experimental setup used for transcriptome analysis. **c** Number of up- and downregulated genes in Col-0 hypocotyls in the indicated light conditions ($FDR < 0.05$, $T$ test with BH correction). **d** GO term enrichment analysis in Col-0 hypocotyl upregulated gene lists. Each node indicates a significantly enriched GO term ($FDR < 0.05$). Two terms (nodes) are connected if they share 20% or more genes. The line thickness increases with the increasing number of shared genes between two terms. Only selected GO terms (black outlines) are annotated. The full list of enriched GO terms is in Supplementary Data 2, the interactive version of (**d**) is available at https://figshare.com/s/864fba30bbd3919d0745. See also Supplementary Fig. 1.

We performed GO enrichment analyses to characterize the processes depending on PIFs and YUCs (Supplementary Data 3). In cotyledons terms related to the biosynthesis of multiple hormones required PIFs but not YUCs (Supplementary Fig. 2c). In contrast, in hypocotyls, terms such as "cell wall organization and biogenesis", "response to brassinosteroids", "response to auxin", and "auxin transport" heavily depended on PIFs and YUCs with a particularly strong dependency for terms related to lipid biosynthesis (Fig. 2c). Given that LRFR YUC-dependent auxin production mostly occurs in cotyledons, this suggests that in hypocotyls LRFR gene induction largely depends on auxin transported from the cotyledons with a potential local action of PIFs.

### The majority of LB-induced genes do not depend on PIFs or YUCs

Based on PCA, *pif457* and *yuc2589* displayed a robust transcriptional response to LB, contrasting with LRFR (particularly in hypocotyls) and the phenotypes of these mutants (Supplementary Fig. 1a and Figs. 1a, 2a, and 3a). Nevertheless, part of the LB-upregulated genes, including some related to protein catabolism and secretion/organelle transport

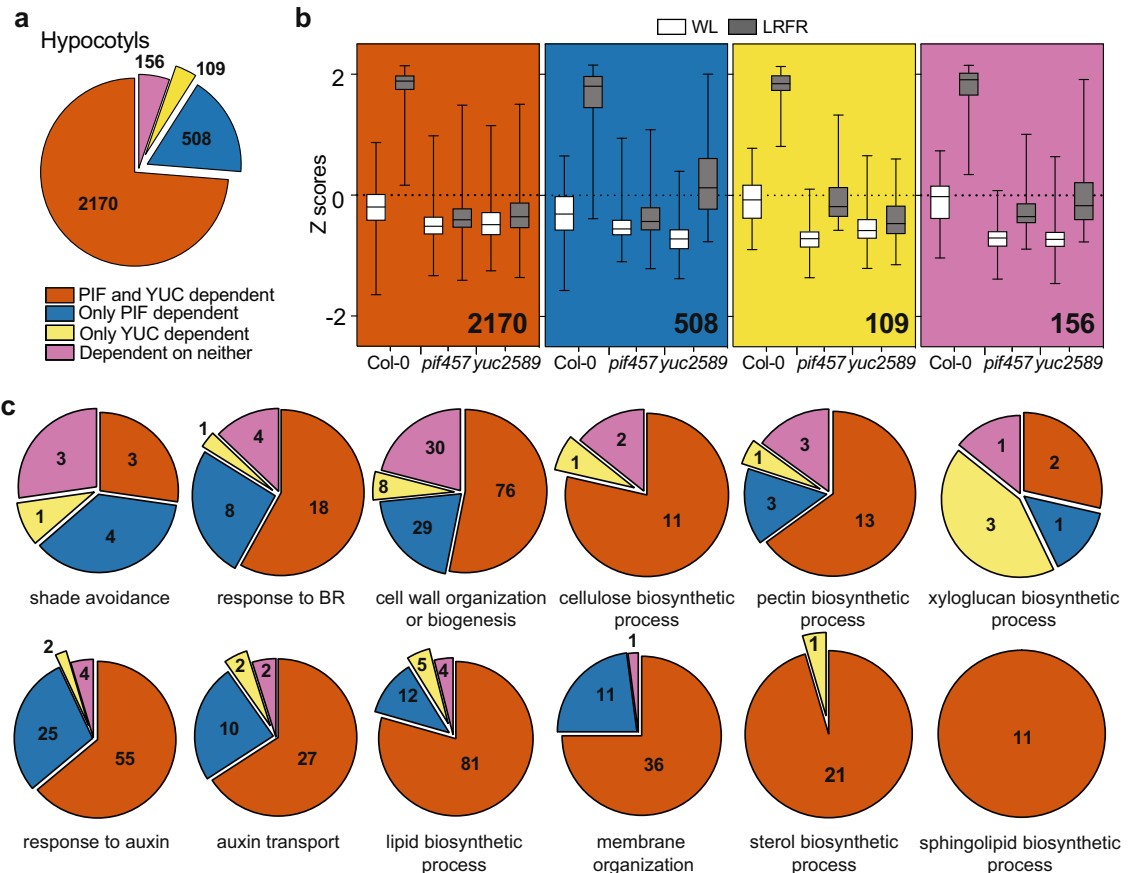

**Fig. 2 | Most LRFR-induced genes in hypocotyls require both PIFs and YUCs.**
**a** The distribution of hypocotyl-induced genes in LRFR according to the dependence on PIFs and YUCs using the comparison of Col-0, *pif457*, and *yuc2589* transcriptomes (FDR < 0.05, *F* test with post hoc test). **b** Distributions of Z-scores computed from replicate averages for categories shown in (**a**). The horizontal bar represents the median; boxes extend from the 25th to the 75th percentile, whiskers extend to show the data range. **c** The distribution of hypocotyl-induced genes according to the dependence on PIFs and YUCs in each of the selected significantly enriched GO terms. Numbers indicate significantly regulated genes in the given categories and/or GO terms. The full list of misregulated genes and enriched GO terms are given in Supplementary Data 3. See also Supplementary Fig. 2.

processes, depended on PIFs and/or YUCs (Fig. 3a, b and Supplementary Fig. 3a, gene and GO term list in Supplementary Data 4). However, the biggest fraction of LB-induced genes did not depend on PIFs and/or YUCs (Fig. 3a, b and Supplementary Fig. 3a). One possibility is that LB-induced hypocotyl elongation requires optimal expression of genes in WL (baseline conditions). Indeed, we found numerous genes with lower expression in *pif457* and/or *yuc2589* compared to WT in WL (Fig. 3c and Supplementary Data 5). We note that most WL-misregulated genes in hypocotyls required both PIFs and YUCs as observed in the hypocotyls of LRFR-treated seedlings (Fig. 3c and Supplementary Fig. 3b). In contrast, in cotyledons more required PIFs but not YUC (Fig. 3c and Supplementary Fig. 3b). Although in the WT LB did not strongly induce these genes, their expression levels in the mutants were also lower in LB (Fig. 3d). Many GO terms related to growth, hormones and cell wall as well as "response to blue light" and "response to starvation" were enriched in the PIF- and YUC-dependent genes in hypocotyls (Supplementary Fig. 3c and Supplementary Data 5). We conclude that in contrast to LRFR, LB-gene induction was less dependent on PIFs and YUCs.

Finally, we compared each set of PIF-dependent genes in WL, LB, and LRFR with putative PIF4 targets[26] (Supplementary Data 6). PIF-dependent genes in WL and LRFR (Figs. 2 and 3c and Supplementary Fig. 3b) were significantly enriched in PIF4 targets for both organs (Fig. 3e and Supplementary Fig. 3d). One exception was PIF and YUC-dependent genes in LRFR in hypocotyls, which may indicate that these genes are induced by auxin produced downstream of PIFs (Fig. 3e).

Similarly, for both organs, promoter motif enrichment analysis showed that PIF-bound sequences (G-box and PBE-box) were overrepresented among PIF-dependent genes in WL and LRFR but not in LB (Fig. 3e, Supplementary Fig. 3d, and Supplementary Data 6). These results suggest that PIFs directly regulate numerous genes in WL conditions, which may contribute to impaired hypocotyl elongation of *pif457* mutants in LB.

### In LRFR PIFs selectively induce *SMT2* expression in the hypocotyl

We confirmed that in hypocotyls LRFR induced numerous genes in diverse anabolic processes (Fig. 1d)[14], predominantly downstream of PIFs and YUCs (Fig. 2c and Supplementary Data 3). The dependency on PIFs could be direct or indirect as PIFs induce YUC-mediated auxin production in cotyledons[13–18]. Previously, we reported that more cotyledon-fixed carbon was allocated into the lipid fraction of elongating hypocotyls in LRFR-treated *B. rapa*[35]. Lipid biosynthesis is required for membrane expansion in rapidly elongating cells[31,32] and was a prominent example of LRFR and PIF-regulated anabolic processes in hypocotyls (Fig. 2c). We focused on sterols because of the global upregulation of the pathway, including several potential direct PIF targets (Figs. 1d and 2c and Supplementary Fig. 4a)[26,45]. Sterols are indispensable constituents of the PM and precursors of BR growth hormones and several biosynthesis mutants are either embryo lethal or show major growth defects[36]. C-24 sterol methyltransferases (SMT), encoded by two paralogs *SMT2* and *SMT3*, act at a branch point leading

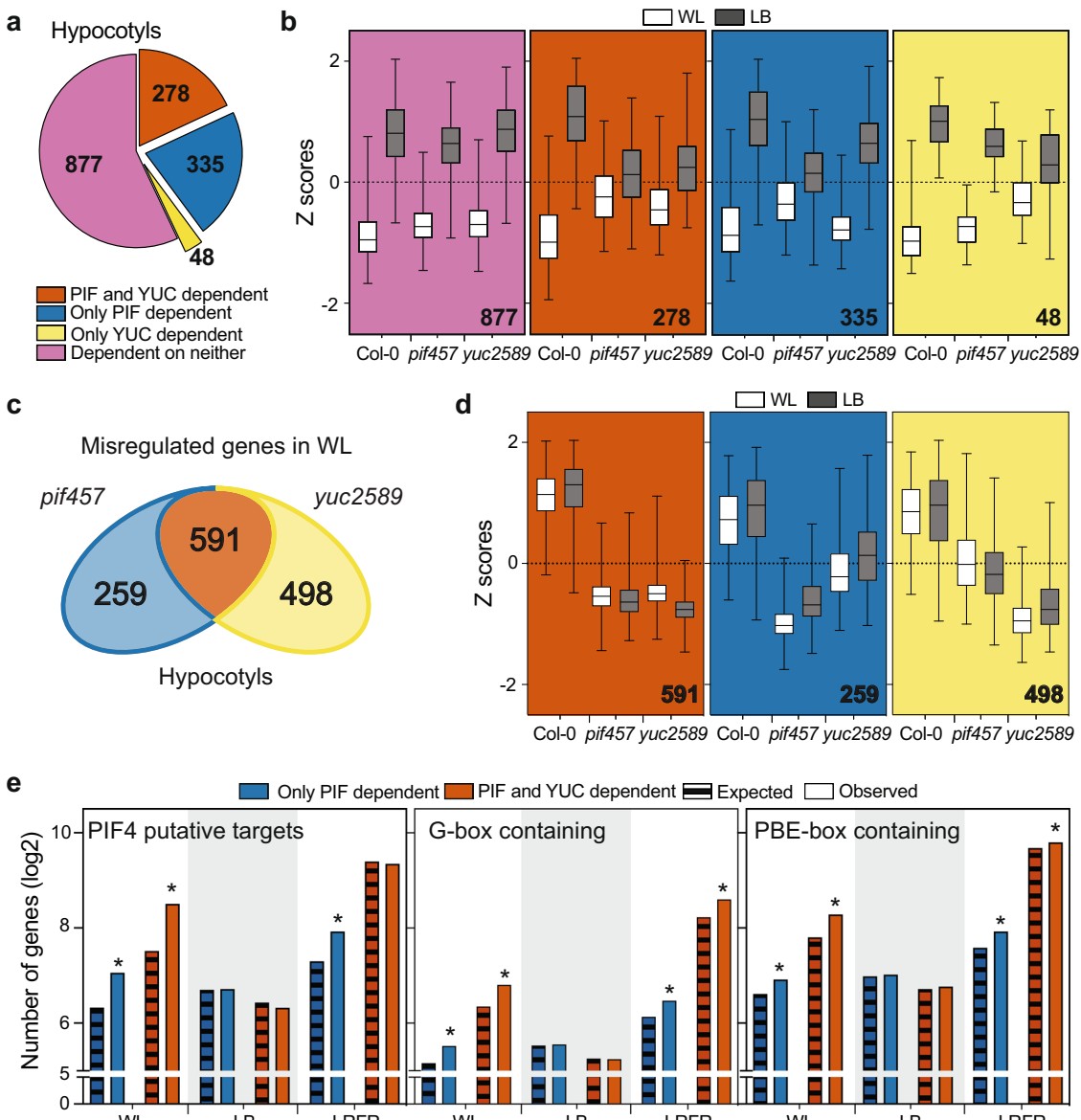

Fig. 3 | **PIFs & YUCs are required for basal expression of many growth- and hormone-associated genes in hypocotyls of WL-grown seedlings. a** The distribution of LB-induced genes in hypocotyls according to the dependence on PIFs and YUCs using the comparison of Col-0, *pif457*, and *yuc2589* transcriptomes (FDR < 0.05, *F* test with post hoc test). **b** Distributions of Z-scores computed from replicate averages for categories shown in (**a**). **c** Number of misregulated genes in *pif457* and *yuc2589* hypocotyls compared to Col-0 in WL (FDR < 0.05, *T* test with BH correction). **d** Distributions of Z-scores computed from replicates averages for genes that are grouped as in (**c**). **e** Comparison of PIF-dependent genes in

hypocotyls with putative PIF4 targets (as listed in ref. 26), promoters (1 kb upstream) containing G-box (CACGTG) or PBE-box (CATGTG). Asterisks (*) indicate the statistically significant overrepresentation compared to expected (*P* < 0.05, Binomial distribution, one-tailed; the exact *P* values are available in the Source Data). **b**, **d** The horizontal bar represents the median; boxes extend from the 25th to the 75th percentile, whiskers extend to show the data range. Numbers indicate significantly regulated genes in the given categories and/or GO terms. The full list of misregulated genes, enriched GO terms, and enriched motifs are given in Supplementary Data 4, 5, and 6. See also Supplementary Fig. 3.

to the synthesis of the predominant PM sterol: sitosterol (Fig. 4a)[33,36,46–48]. Sitosterol levels decrease dramatically in *smt2* and marginally in *smt3* mutants but these mutants still contain high levels of other sterols (e.g., campesterol) and BR and they do not have serious growth defects[46–48]. This enabled us to conduct physiological and molecular experiments using these mutants.

Induction of *SMT2* and *SMT3* expression specifically occurred in LRFR, selectively in hypocotyls, and in a PIF and YUC-dependent manner (Fig. 4b). Furthermore, LRFR led to enhanced PIF4-HA binding to the promoter of *SMT2* and *SMT3* (Fig. 4c) in *PIF4p:PIF4-HA* (*pif4-101*) seedlings[49]. We also detected a significant PIF7-HA enrichment on the *SMT3* but not the *SMT2* promoter (Supplementary Fig. 4b) in *PIF7p:-PIF7-HA* (*pif7-2*)[50]. PIF7-HA binding to *SMT3* and *HFR1*, the latter being

used as a positive control, was much lower than PIF4-HA binding (Fig. 4c and Supplementary Fig. 4b). Taken together, our data show that PIFs induce *SMT2* and *SMT3* expression specifically in hypocotyls and this is accompanied by enhanced PIF binding to *SMT* promoters in LRFR.

We used *smt2* and *smt3* mutants[46–48] to test the importance of these genes in hypocotyl growth. Hypocotyl elongation was significantly reduced in two independent *smt2* alleles and *smt2smt3* double mutants, whereas the response to LRFR was unaffected in *smt3* (Fig. 4d and Supplementary Fig. 4c). Similarly, when applied simultaneously with light treatments, fenpropimorph, which inhibits sterol biosynthesis upstream of SMTs[33,51], reduced hypocotyl elongation in LRFR (Fig. 4a, e). Furthermore, increased drug concentration resulted

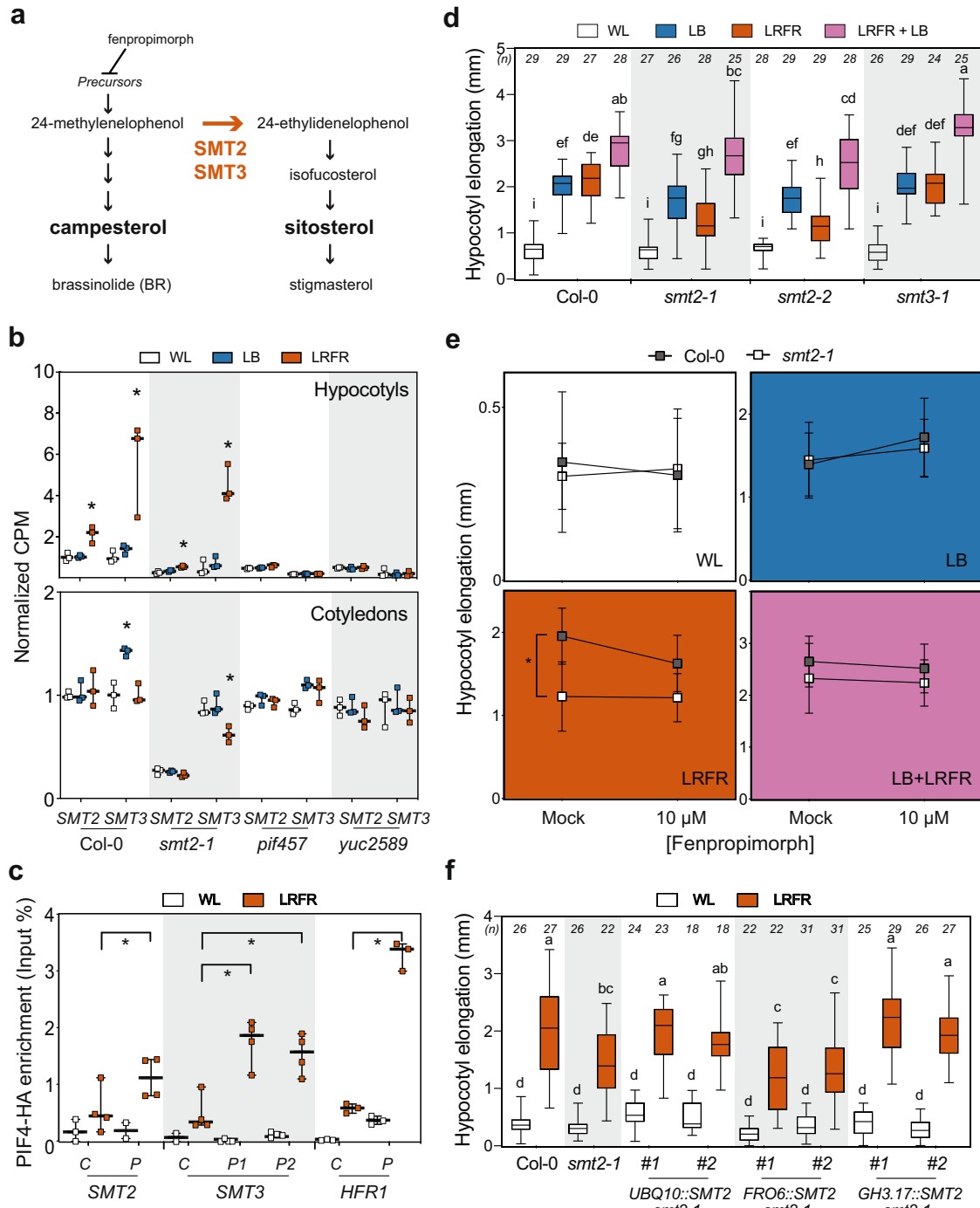

**Fig. 4 | SMT2 is required locally for LRFR-induced hypocotyl elongation. a** A simplified representation of sterol biosynthesis pathway in *Arabidopsis*[46]. **b** Normalized CPM (counts per million, normalized to Col-0 average in WL) of *SMT2* and *SMT3* in the indicated genotypes and conditions (data from RNA-seq). **c** PIF4-HA binding to the promoter of the indicated genes. PIF4-HA enrichment is quantified by qPCR and presented as IP/Input. Control−C, Peak−P. **d**–**f** Hypocotyl elongation of the indicated genotypes. **d**, **f** The horizontal bar represents the median; boxes extend from the 25th to the 75th percentile, whiskers extend to show the data range. **b**, **c** Each data point indicates biologically (**b**) or technically (**c**) independent samples, the horizontal bar represents the median, whiskers extend to show the data range. **e** Data are means ± SD with a regression line. Different letters

(**d**, **f**, two-way ANOVA with Tukey's HSD test) and asterisks (*) (**b**, **c**, Student's *T* test, one-tailed) (**e**, two-way ANOVA) indicate a significant difference ($P < 0.05$) compared to WL (**b**) or control (**c**), and between genotypes in given light condition (**e**). The exact *P* values are available in the Source Data. Sample size (*n*) that is given on top (**d**, **f**); **e** for Col-0; *n* (WL-Mock) = 21, *n* (WL-10 μM) = 21, *n* (LB-Mock) = 26, *n* (LB-10 μM) = 26, *n* (LRFR-Mock) = 24, *n* (LRFR-10 μM) = 24, *n* (LB + LRFR-Mock) = 25, *n* (LB + LRFR-10 μM) = 25; for *smt2-1*; *n* (WL-Mock) = 12, *n* (WL-10 μM) = 14, *n* (LB-Mock) = 20, *n* (LB-10 μM) = 21, *n* (LRFR-Mock) = 16, *n* (LRFR-10 μM) = 18, *n* (LB + LRFR-Mock) = 24, *n* (LB + LRFR-10 μM) = 18) indicates biologically independent seedlings examined over one experiment. The experiments were repeated two (**e**) and three (**c**, **d**, **f**) times with similar results. See also Supplementary Fig. 4.

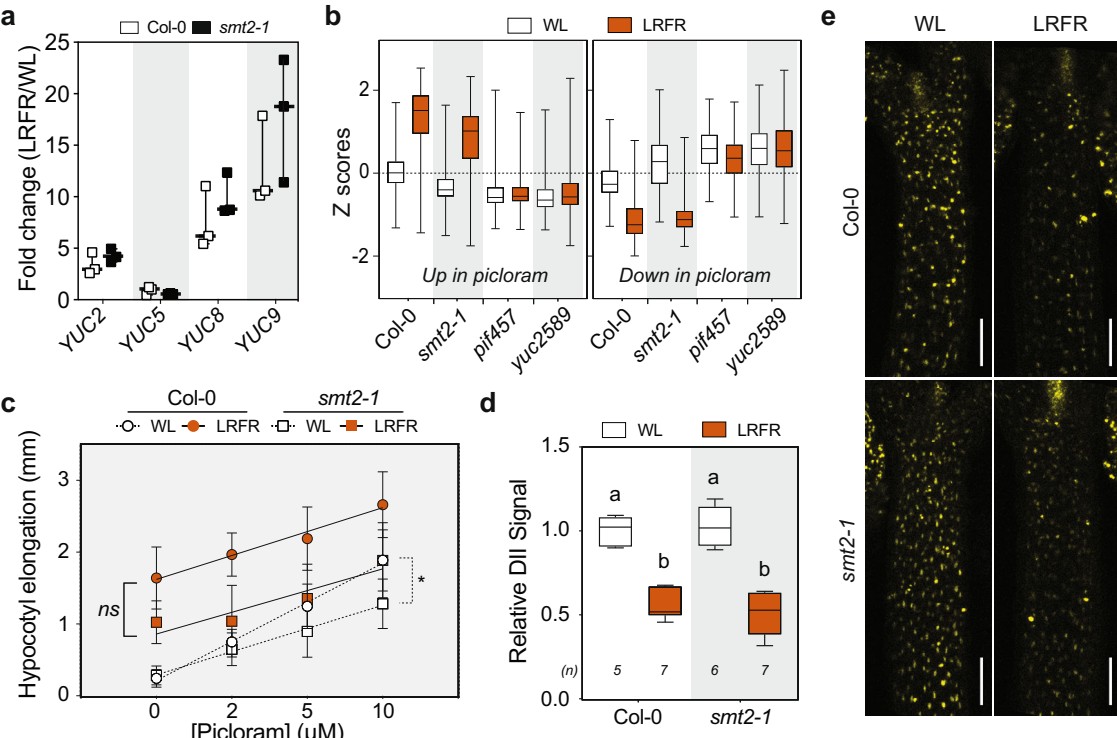

**Fig. 5 | Auxin biosynthesis, response, and transport are normal in *smt2-1* in LRFR. a** LRFR-induction of auxin biosynthetic genes in the indicated genotypes (data from RNA-seq). Each data point indicates biologically independent samples, the horizontal bar represents the median, whiskers extend to show the data range. **b** Distributions of Z-scores computed from replicates averages for synthetic auxin picloram up- and downregulated genes in hypocotyls of the indicated genotypes (as listed in ref. 55). **c** Hypocotyl elongation of indicated genotypes with the indicated doses of picloram. Data are means ± SD with a regression line. **d**, **e** Quantification (**d**) and the representative images (**e**) of the DII-VENUS signal intensity (normalized to mean value of Col-0 in WL) in hypocotyls of the indicated genotypes either kept at WL or transferred to LRFR for 1 h. White bars equal to 100 μm. **b**, **d** The horizontal bar represents the

median; boxes extend from the 25th to the 75th percentile, whiskers extend to show the data range. Different letters (**d**, two-way ANOVA with Tukey's HSD test) and asterisks (*) (**c**, two-way ANOVA) indicate a significant difference ($P < 0.05$) between genotypes in given light condition (**c**) and compared to WL (**d**). The exact $P$ values are available in the Source Data. Sample size ($n$) that is given on top (**d**); ((**c**) For Col-0: $n$ (WL) = 17, 22, 21, 20; $n$ (LRFR) = 19, 22, 17, 22. For *smt2-1*: $n$ (WL) = 12, 15, 19, 17; $n$ (LRFR) = 13, 18, 17, 17 with an order of Mock, 2 μM, 5 μM, 10 μM respectively) indicates biologically independent seedlings examined over one experiment. The experiments were repeated two (**c**) and three (**d**) times with similar results. The full gene list of LRFR-picloram transcriptome comparison is given in Supplementary Data 8. See also Supplementary Fig. 5.

in a steeper reduction in hypocotyl elongation of WT compared to *smt2-1* in LRFR (Supplementary Fig. 4d). Interestingly, hypocotyl elongation of *smt2* and *smt3* mutants was as in the WT in WL and LB (Fig. 4d). These phenotypes highlight the stronger requirement for *SMT2/SMT3* in LRFR than in other light conditions (Fig. 4b, d). Remarkably, LB + LRFR combination mimicking vegetative shade rescued the reduced hypocotyl elongation of *smt2-1* in LRFR (Fig. 4d). Similarly, inhibition of sterol biosynthesis during LB and LB + LRFR treatments did not reduce hypocotyl elongation (Fig. 4e). This result contrasts with the hypocotyl phenotype of *smt2smt3* in LB and LB + LRFR (Supplementary Fig. 4c). The *smt2smt3* double mutant contains only trace amounts of sitosterol, whereas sitosterol biosynthesis is partially impaired in *smt2* and upon fenpropimorph treatment (at the tested concentration)[46,48,51]. Thus, these results indicate that in the *smt2* mutant, which contains low levels sitosterol, hypocotyl elongation is primarily affected in LRFR while in the *smt2smt3* double mutant, which contains trace amounts of sitosterol, hypocotyl elongation is impaired more broadly.

To determine whether SMT2 is required locally for LRFR-induced hypocotyl growth, we phenotyped *smt2-1* transformed with the *SMT2* coding sequence controlled by a ubiquitous (*UBQ10*), cotyledons specific (*FRO6*)[52] or hypocotyls-specific promoter (*GH3.17*)[53] (Supplementary Fig. 4e). *UBQ10*- and *GH3.17*-driven *SMT2* rescued the *smt2-1* phenotype in LRFR in two independent insertion lines for each construct, whereas *FRO6* did not (Fig. 4f). Taken together our data shows

that PIF selectively regulate *SMT2/SMT3* expression in the hypocotyl (Fig. 4b, c) and that *SMT2* expression in the hypocotyl is functionally important (Fig. 4f).

One characteristic phenotype of the *smt2* and *smt2smt3* mutants is an impaired cotyledon vasculature pattern (cvp) (Supplementary Fig. 5a)[46–48]. Auxin transport from cotyledons to hypocotyls is required for LRFR-induced elongation[19,20,28]. Thus, we determined hypocotyl growth of other severe *cvp* mutants, *cvp2* and *cvp2cvl1* (Supplementary Fig. 5a) that do not interfere with sterol biosynthesis[54]. Both *cvp* mutants showed normal hypocotyl elongation in LRFR (Supplementary Fig. 5b), suggesting that the cotyledon vasculature problem of *smt2* alone does not explain its hypocotyl growth defect in LRFR. To further analyze the *smt2* phenotype, we used RNA sequencing, which showed that the transcriptome of *smt2-1* was similar to Col-0 in LB and LRFR in both organs (Supplementary Fig. 5c, d and Supplementary Data 7). LRFR-induced expression of the major genes coding for auxin biosynthesis was similar in Col-0 and *smt2-1* (Fig. 5a). Furthermore, comparison of our LRFR transcriptome and 2 h picloram (synthetic auxin)-regulated genes in hypocotyls[55] showed a high correlation between these treatments for Col-0 and *smt2-1* (Fig. 5b and Supplementary Data 8), but not in *pif457* and *yuc2589* which are impaired in auxin biosynthesis in LRFR[14,18,56]. The picloram dose response of WT and *smt2-1* hypocotyls was similar in LRFR (Fig. 5c), indicating a similar auxin response. Finally, DII-VENUS, an auxin-input reporter[57], similarly decreased in hypocotyls of LRFR-treated seedlings of both genotypes,

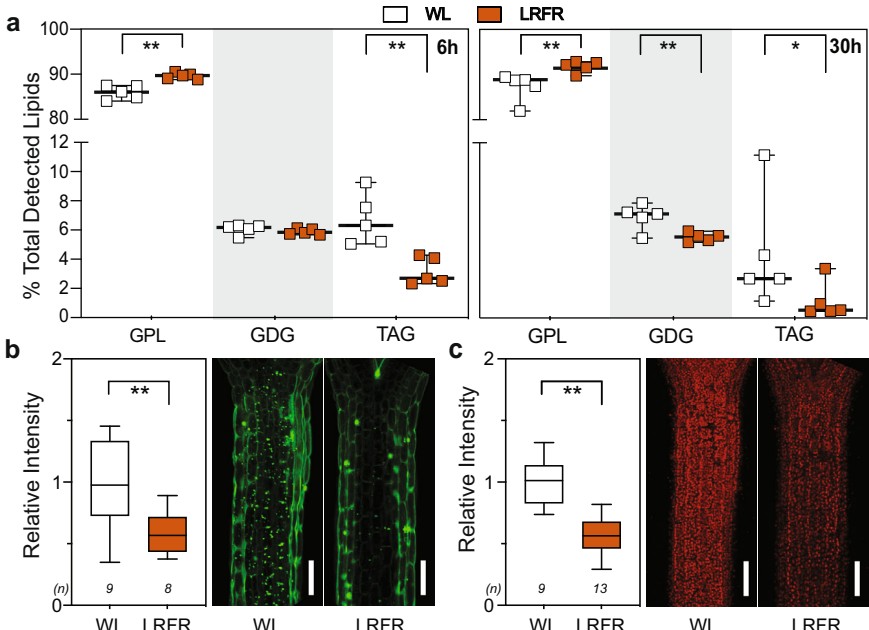

**Fig. 6 | LRFR changes the lipid composition in hypocotyls. a** Lipid class abundance at the indicated time points is represented as percentage of total detected lipids in *B. rapa* hypocotyls. Each data point indicates biologically independent samples, horizontal bar represents the median, whiskers extend to show the data range. GPL glycerophospholipids, GDG glycosyldiacylglycerols, TAG triacylglycerols. **b, c** Quantification (left) and representative images (right) of (**b**) LDs using BODIPY™ 493/503 and (**c**) chloroplasts in Col-0 hypocotyls either kept at WL or transferred to LRFR for 30 h. Data is normalized to WL average. White bars are scale bars of 100 μm. **b, c** The horizontal bar represents the median; boxes extend from the 25th to the 75th percentile, whiskers extend to show the data range. **a–c** Asterisks indicate *P* values (*<0.1, **<0.05, Student's *T* test, one-tailed). The exact *P* values are available in the Source Data. **b, c** Sample size (*n*) that is given on top indicates biologically independent seedlings examined over one experiment. The experiments were repeated three times with similar results. The full list of detected lipid species is given in Supplementary Data 9. See also Supplementary Fig. 6.

indicating increased auxin levels (Fig. 5d, e). Altogether, our data suggest that reduced *smt2* hypocotyl elongation in LRFR unlikely results from major alterations in auxin biosynthesis, transport or response.

## Regulation of the lipid composition in the hypocotyls of LRFR-treated seedlings

We next analyzed the total lipid content in LRFR using untargeted lipidomics[58] in *B. rapa* hypocotyls, which respond to LRFR similarly to Arabidopsis and allocate more newly fixed carbon to lipids[35]. This choice was also dictated by technical limitations rendering this experiment impossible with dissected Arabidopsis hypocotyls. The percentage of major PM lipids (glycerophospholipids–GPL) increased, whereas the storage lipids (triacylglycerols–TAG) and the major constituents of thylakoid membranes (glycosyldiacylglycerols–GDG)[33] decreased significantly in LRFR in the total lipid pool (Fig. 6a). We then tested whether similar adjustment of lipid profiles occurred in Arabidopsis hypocotyls. We used BODIPY™ 493/503 that stains neutral lipids[59] to detect the level of lipid droplets (LD) that mainly contain TAG[60]. Fluorescence intensity of LDs decreased significantly after 30 h of LRFR treatment in Arabidopsis hypocotyls (Fig. 6b). Furthermore, our transcriptome data showed that the "thylakoid membrane organization" GO term was enriched among hypocotyl downregulated genes in LRFR (Supplementary Data 3). In line with this data and previous reports on tomato stems[61], the fluorescence intensity of chloroplasts decreased in Arabidopsis hypocotyls in LRFR (Fig. 6c). Our data indicate that LRFR alters the lipid composition in the hypocotyls of *B. rapa* and Arabidopsis.

Although we focused on *SMT2* and *SMT3*, LRFR led to a coordinated induction of sterol biosynthesis genes in Arabidopsis hypocotyls (Supplementary Fig. 4a)[14]. This suggests that LRFR induces a general increase in sterols. Thus, we determined the sterol composition in *B. rapa* hypocotyls where LRFR-expression profiles of *BrSMTs* were

similar to their orthologs in Arabidopsis (Supplementary Fig. 6a, *BrIAA29* being a control)[20]. Campesterol and sitosterol, both major sterols in the PM[36], did not change in LRFR during the timeframe of our experiments (Supplementary Fig. 6b). Yet, the percentage of ergosta-5,7-dienol, a precursor for BRs downstream of campesterol, decreased after 3 h of LRFR (Supplementary Fig. 6b). This is in line with a report showing a decreased level of another BR precursor in LRFR[62]. These results suggest that LRFR induces a total increase in sterols rather than a major change in their composition. We also analyzed plant Sphingolipids using a dedicated LC-MS[2] protocol as the methods used for untargeted lipid analysis are poorly suited to study this important class of PM lipids[63]. As expected, this analysis showed that GIPC (Glycosyl Inositol Phospho Ceramides) were the most abundant class of Sphingolipids in *B. rapa* hypocotyls and did not reveal any LRFR-induced change within this lipid class (Supplementary Fig. 6c).

## LB induces autophagy

In our LB treatment, PAR was reduced to 66% of WL levels while it remained unchanged in LRFR (Supplementary Fig. 7a). Such a decrease in PAR results in around 50% reduction in the net $CO_2$ assimilation, regardless of the light color used for illumination (B, G, R, or their combinations)[2]. In contrast, carbon fixation remained unchanged in *B. rapa* seedlings in LRFR[35]. Consistently, the "carbon fixation" GO term was enriched only among LB downregulated genes, while terms related to carbon starvation were enriched in LB-induced genes (Fig. 1d, Supplementary Fig. 8a, and Supplementary Data 3, 4). These data suggest that LB induced a switch to a catabolic state to enable growth with declining carbon availability. Accordingly, we observed the selective enrichment of catabolism terms and "Autophagy" in LB-induced genes (Figs. 1d and 7a). To test whether LB induces autophagy, we used a ubiquitously expressed *GFP-ATG8a* line to quantify autophagic flux[64–67]. As free GFP is more resistant to vacuolar degradation than GFP-ATG8a, bulk autophagy results in the accumulation of free

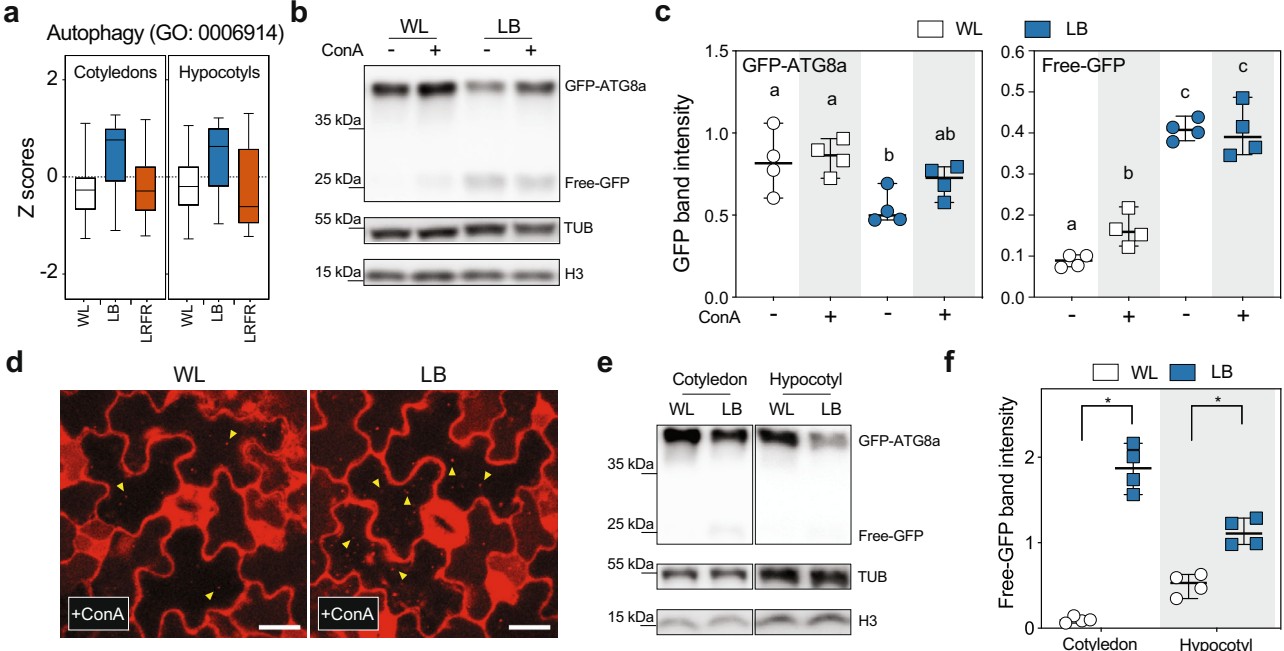

**Fig. 7 | LB induces autophagy. a** Distributions of Z-scores computed from replicates averages for genes listed in autophagy GO term in Col-0 seedlings. The horizontal bar represents the median; boxes extend from the 25th to the 75th percentile, whiskers extend to show the data range. The full list of genes in the Autophagy GO term is given in Supplementary Data 10. **b** Representative image of autophagic flux assay using *35 S:GFP-ATG8a* (WT), seedlings were grown in the indicated light conditions with or without ConA (0.5 μM). GFP-ATG8a and free GFP levels were detected in total protein extracts with an anti-GFP antibody. **c** Quantification of GFP-ATG8a and free-GFP bands in autophagic flux assays. Different letters indicate a significant difference (*P* < 0.05, two-way ANOVA with Tukey's HSD test; the exact *P* values are available in the Source Data). **d** Representative image of cotyledon pavement cells expressing *UBQ10:mCherry-ATG8e* in the WT background either kept in WL or treated with

LB for 8 h in the presence of concanamycin A (ConA, 5 μM). Yellow arrowheads indicate autophagic bodies. White bars equal to 20 μm. The experiment was repeated two times with similar results using 5 to 7 biologically individual seedlings for each condition and the experiment. **e** Western blot analysis of GFP-ATG8a and free-GFP levels in cotyledons and hypocotyls of dissected WT seedlings treated with 8 h of LB without ConA. **f** Quantification of free-GFP bands in (**e**). Asterisks indicate *P* values (*<0.001, Student's *T* test, one-tailed; the exact *P* values are available in the Source Data). **b**, **e** H3 and TUB were used as loading control. The experiment was repeated four times with similar results. **c**, **f** Each data point indicates a biologically independent sample (average of two technical replicates for each data point), horizontal bar represents the median, and whiskers extend to show the data range. See also Supplementary Fig. 8.

GFP in the vacuole[68]. We showed that free GFP intensity increased in LB, whereas the full-length GFP-ATG8a decreased in the absence of a vacuolar-type v-ATPase inhibitor, concanamycin A (ConA) (Fig. 7b, c), indicating LB-induced autophagic flux. Furthermore, GFP-ATG8a levels did not decrease significantly upon ConA application, indicating that LB triggered vacuolar degradation of autophagic bodies. As controls, we showed that GFP levels remained unchanged in a *35 S:GFP* line in LB and that in *atg7 GFP-ATG8a* LB did not lead to an increase in free GFP (Supplementary Fig. 8b, c). Since ATG8 is found in autophagosomal membranes[69], we used a ubiquitously expressed *mCherry-ATG8e* line[70] to monitor autophagic activity microscopically using an independent assay. The number of autophagic bodies increased in LB-treated cotyledons in the presence of ConA (Fig. 7d and Supplementary Fig. 8d). We did not detect autophagic bodies in the *atg5-1* autophagy-deficient mutant[64,67], in LB confirming the nature of these structures (Supplementary Fig. 8e). To test whether LB induces autophagy in hypocotyls and cotyledons, we dissected seedlings and used the GFP-ATG8a western blot assays, which showed that LB led to a significant increase in free GFP in both organs (Fig. 7e, f). The microscopic assay in hypocotyls was not possible presumably due to poor penetration of ConA in this tissue. Altogether, our data indicate that LB promotes autophagy in hypocotyls and cotyledons transcriptionally and by using autophagy reporters.

To determine what light feature of LB triggers autophagy, we used the GFP-ATG8a western blot assays to compare WL with LB and WL with low PAR (LP) corresponding to PAR in the LB treatment (66% of WL). This experiment showed that LB but not LP led to an increase in

free GFP (Fig. 8a, b). Moreover, we determined how these treatments affect hypocotyl elongation also including a LB treatment with increased PAR corresponding to PAR in WL. This experiment showed that LB (high or low PAR) but not LP induced hypocotyl elongation (Fig. 8c). We next determined whether autophagy is required for LB-induced hypocotyl elongation using the autophagy mutants *atg7-2* and *atg5-1*[64,71] (Fig. 8e and Supplementary Fig. 9a). Both *atg7-2* and *atg5-1* hypocotyls elongated less than the WT in all tested light conditions except in WL, with a stronger reduction in LB than LRFR (Fig. 8e and Supplementary Fig. 9a). Remarkably, the *atg7* hypocotyl phenotype contrasts with *smt2-1*, which had the strongest phenotype in LRFR (Fig. 8e). Moreover, LB + LRFR significantly enhanced hypocotyl elongation in *atg5*, *atg7* and *smt2-1* single mutants suggesting a degree of compensation between anabolic and catabolic processes mainly promoted by LRFR and LB, respectively (Fig. 8e). Supporting this idea, *smt2atg7* hypocotyls elongated neither in LB nor in LRFR (Fig. 8e). Importantly, *smt2atg7* hypocotyl elongated marginally in LB + LRFR (simulated vegetative shade), a light condition that induced autophagy similarly to LB (Fig. 8d, e). We further confirmed that these phenotypes are specific to LB, showing that LP did not induce hypocotyl elongation in any of these mutants (Supplementary Fig. 9b, c) Finally, we tested whether the LB hypocotyl growth phenotype of *atg7* could be rescued by applying exogenous sucrose and found that *atg7* hypocotyl growth was significantly induced by sucrose compared to sorbitol that is used as an osmotic control (Supplementary Fig. 9d). Thus, we conclude that LB and not an equivalent reduction in PAR (without affecting the light spectrum) induces both autophagy and hypocotyl elongation.

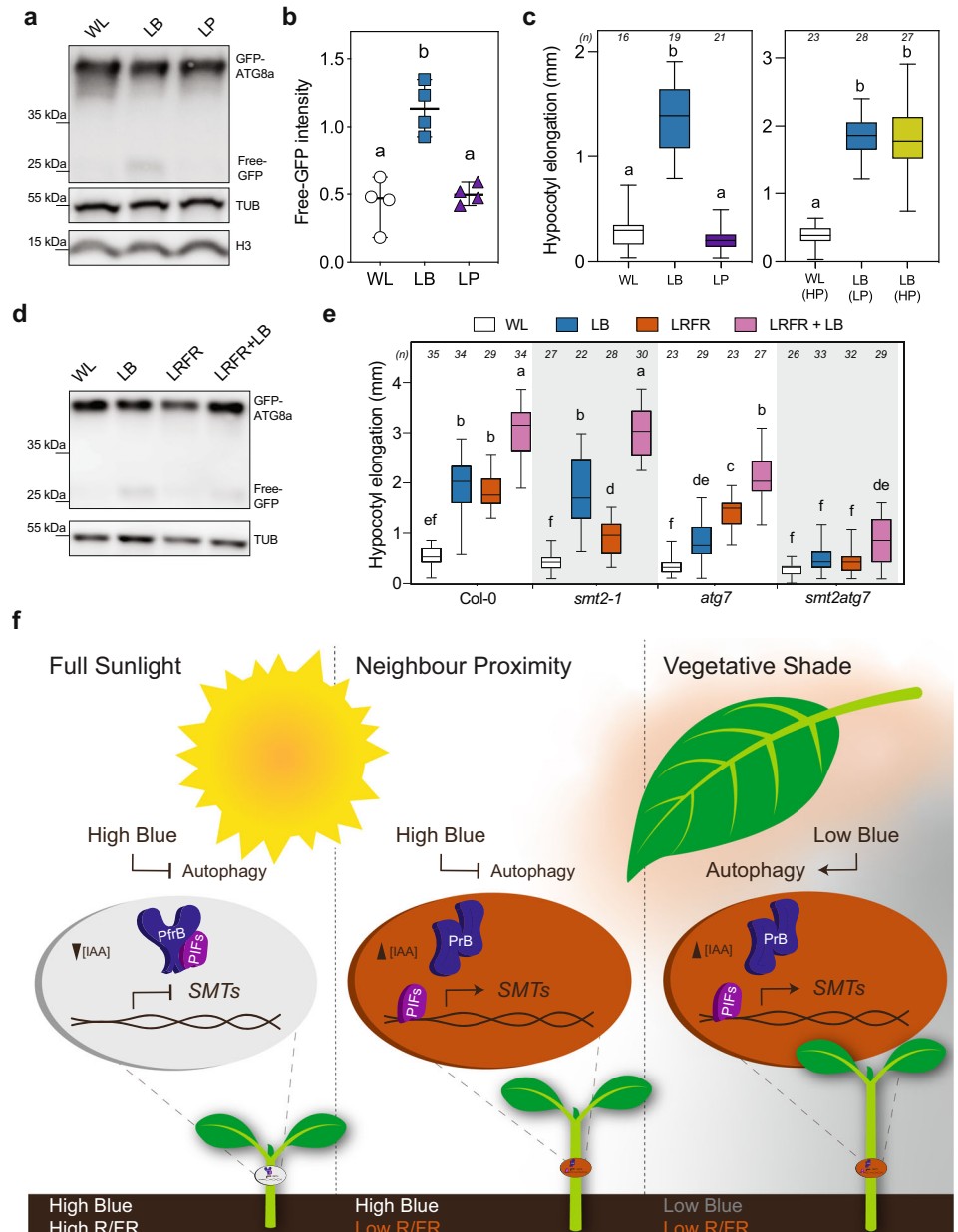

**Fig. 8 | Hypocotyl elongation phenotypes correlate well with induction of autophagy in response to changing light environments. a** Western blot analysis of GFP-ATG8a and free GFP in WT seedlings treated for 8 h with LB or LP without ConA. **b** Quantification of free-GFP bands in (**a**). Each data point indicates a biologically independent sample (average of two technical replicates for each data point), horizontal bar represents the median, and whiskers extend to show the data range. **c** Hypocotyl elongation of WT seedlings treated with LB, LP, or LB with increased PAR (High PAR, HP) compared to WL. **d** GFP-ATG8a and free-GFP levels are detected in WT seedlings treated with 8 h of LB, LRFR, or LRFR + LB without ConA. **e** Hypocotyl elongation of seedlings with the indicated genotypes treated with the indicated light conditions. The sample size (*n*) that is given on top indicates biologically independent seedlings examined over one experiment. The

experiments were repeated four (**a**, **d**) and three (**e**) times with similar results. H3 (**a**, **d**) and TUB (**d**) were used as a loading control. **c**, **e** The horizontal bar represents the median; boxes extend from the 25th to the 75th percentile, whiskers extend to show the data range. **b**, **c**, **e** Different letters indicate significant difference ($P < 0.05$, two-way ANOVA with Tukey's HSD test, the exact $P$ values are available in the Source Data). **f** Model for hypocotyl elongation in full sunlight, neighbor proximity, and vegetative shade conditions. In full sunlight (left panel) and neighbor proximity (middle panel), plants receive high-intensity blue light (high blue, shown by the color white) and autophagy remains at basal levels. Vegetative shade (right panel) decreases the blue light intensity (low blue, shown by the dark-gray color) and promotes autophagy-mediated recycling. The reduced R/FR ratio in neighbor proximity and vegetative shade inactivates phyB thereby promoting PIF activity.

Moreover, autophagy is important to promote hypocotyl elongation in a vegetative shade that combines LB and LRFR (Fig. 8e, f).

## Discussion

In young seedlings, cotyledons are the major organs sensing LRFR, while growth promotion occurs in hypocotyls[14,20,72]. In the cotyledons, the PIF-YUC regulon controls the production of auxin that promotes elongation upon transport to the hypocotyl. However, how PIFs

control hypocotyl elongation locally (in the growing organ) and how PIF activity may be induced in hypocotyls is less clear. Modulation of auxin sensitivity in hypocotyls is one identified mechanism[10,13,14,73] (Fig. 2c). Here, we show organ-specific transcriptional control of *SMT2* and *SMT3* by PIFs (Fig. 4b). ChIP data indicate that PIF4 and PIF7 directly control expression of those genes with enhanced binding to their promoter in LRFR (Figs. 4c and 8 and Supplementary Fig. 4b). LRFR-induced *SMT2* and *SMT3* expression also depends on

YUC-mediated auxin production (Fig. 4b), yet auxin alone does not induce their expression in hypocotyls[55]. This suggests a combined function of PIFs and auxin for LRFR-induced expression of *SMT2* and *SMT3* (Fig. 8f). Similarly, most LRFR-induced genes in the hypocotyl belonging to GO categories related to growth-promoting processes depend on both PIFs and YUCs (Fig. 2 and Supplementary Data 3). This regulatory pattern suggests that in hypocotyls, increased auxin levels may promote PIF-mediated gene expression. This hypothesis is consistent with coordinated regulation of growth-regulatory genes by ARF6, BZR1, and PIF4[21]. Furthermore, several *PIF*s are putative targets for ARF6 and BZR1[21]. BZR1 is also known to induce *PIF4* expression during thermomorphogenesis[74] that induces hypocotyl elongation similarly to LRFR downstream of PIF4 and PIF7[45,75,76]. In contrast to hypocotyls, a substantial fraction of LRFR-induced genes in cotyledons depend on PIFs but not YUCs (Supplementary Fig. 2a). This is consistent with earlier studies identifying several LRFR-induced genes that do not depend on de novo auxin production[77]. A change in light conditions in cotyledons leads to direct PIF activation potentially explaining these findings. Collectively, our data reveal organ-specific patterns of PIF-mediated gene induction in hypocotyls versus cotyledons and identify *SMT2* as an example of a gene that is selectively induced in the hypocotyl and is particularly important during LRFR-induced hypocotyl elongation.

LRFR leads to enhanced reallocation of newly fixed carbon to the lipid fraction of *B. rapa* hypocotyls[35]. The expression of many lipid-biosynthetic genes, including sterols is induced in the hypocotyl of LRFR-treated Arabidopsis seedlings (Fig. 1). In addition to sterols, LRFR upregulates the biosynthetic genes for other major components of PM lipids (e.g., sphingolipids) (Fig. 2c)[33]. PM extension depends on the deposition of lipids that occurs during the delivery of membranes via exocytosis[31–34]. Of note, endocytosis and exocytosis are also GO terms enriched among upregulated genes in LRFR (Fig. 1d). On the other hand, terms related to chloroplast lipids (GDG) are found among downregulated genes in LRFR (Supplementary Data 3). Consistent with the transcriptional data in Arabidopsis, GPL increases while TAG decreases in *B. rapa* hypocotyls (Fig. 6a). A similar decrease in chloroplasts was previously observed in tomato stems in LRFR[61]. Moreover, we show that in Arabidopsis hypocotyls both chloroplasts and storage lipids decrease in LRFR (Fig. 6b, c). Interestingly, during de-etiolation phytochromes also control storage lipid utilization[78]. Overall, our data indicate that LRFR leads to enhanced production of sterols with a reduction of non-PM lipid classes (Fig. 6a). SMT2 and SMT3 act at a branch point of sterol biosynthesis (Fig. 4a). Their induction by LRFR may therefore lead to changes in sterol composition which is potentially important given their role in PM fluidity and microdomain organization[33,36]. We cannot rule out this possibility, however our sterol and sphingolipid measurements in LRFR-treated *B. rapa* do not provide evidence for such a change (Supplementary Fig. 6). We also note that coordinated transcriptional upregulation of the sterol pathway (Supplementary Fig. 4a) is consistent with enhanced overall sterol demand rather than indicative of changes in sterol composition.

PIFs and auxin production are also functionally important for LB-induced hypocotyl elongation[24–26,28] (Fig. 1a). However, robust LB-regulated gene expression still occurred in *pif457* (Supplementary Fig. 1a). Furthermore, many PIF-dependent genes in LB are probably not direct PIF targets (Fig. 3e, Supplementary Fig. 3d, and Supplementary Data 6). Interestingly, many auxin, BR, and GA response and other growth-related genes show reduced basal (WL) expression in *pif457*, and this difference persists in LB (Fig. 3 and Supplementary Data 5). Importantly, mutant analysis and pharmacological treatments show an indispensable role for auxin, BR, and GA during LB-induced hypocotyl elongation (Fig. 1a)[25,26,28,29]. We thus conclude that a more general gene expression defect (already present in WL) may contribute to the hypocotyl growth defect of *pif457* mutants in LB. This is consistent with a report showing that PIF4 and PIF5 are dose-dependent

inducers of hypocotyl elongation in WL[44]. However, hypocotyl elongation and PIF4 and PIF5 accumulation occurs more slowly in LB than LRFR[26,44,73,79]. Hence, our analysis at 3 h may have missed some of the PIF-regulated transcriptional events. Therefore, the link between PIF-regulated gene expression and hypocotyl growth control in LB requires further investigation.

While LRFR leads to transcriptional changes in the hypocotyl, possibly indicative of enhanced production of building blocks required for growth, LB leads to the induction of many catabolic processes and autophagy-related genes (Fig. 1d and Supplementary Fig. 1c). The LRFR gene expression pattern suggests that when light resources remain available (Supplementary Fig. 7a), the Target of Rapamycin (TOR) pathway is on and promotes e.g., the biogenesis of ribosomes and nucleotides[80]. The rapid and concomitant rise in auxin and many anabolic processes in hypocotyls (Fig. 1d and Supplementary Fig. 1c)[14] suggest a potential link between these processes. Consistently, a recent report shows that sugar-dependent TOR activity requires auxin signaling[81]. Unlike LRFR, LB leads to PAR reduction (Supplementary Fig. 7a). LB presumably induces metabolic adjustments including alternative pathways for respiration (e.g., protein catabolism) to sustain growth, which is consistent with the situation in other carbon limiting conditions[82–84]. LB transcriptionally promotes autophagy, leads to enhanced production of autophagic bodies (visualized with mCherry-ATG8e) and autophagic processing of GFP-ATG8a in hypocotyls and cotyledons, while LRFR alone does not (Figs. 7 and 8). Importantly, the reduction in PAR alone (maintaining the same light spectrum) to the level in our LB treatment neither induces hypocotyl elongation nor autophagy (Fig. 8 and Supplementary Fig. 9). This indicates that LB rather than the reduction in PAR induces autophagy and hypocotyl elongation. Consistently, *atg7* has a hypocotyl growth defect in LB but not WL or LP (Fig. 8 and Supplementary Fig. 9). Interestingly, the combined LB + LRFR treatments, which mimic vegetative shade, largely rescues the phenotypes of *atg7* and *smt2* single mutants, suggesting a degree of compensation between catabolic and anabolic processes. Consistently, *smt2 atg7* double mutants did not elongate in LB or LRFR and had a modest growth response in combined treatments (Fig. 8e). Importantly, *SMT2* and *SMT3* expression is induced in the hypocotyl of shade-treated seedlings (combined LB and LRFR)[85]. Altogether, our work indicates a particularly important requirement for enhanced de novo synthesis in LRFR and autophagy in LB, while the combination of both processes contributes to growth enhancement of the hypocotyl in vegetative shade.

In conclusion, our work shows that the mechanisms underlying hypocotyl growth promotion differ during neighbor proximity (LRFR) and in vegetational shade that comprises both LB and LRFR. These differences are illustrated in a model (Fig. 8f). We note that thermomorphogenesis and neighbor proximity both lead to similar growth adaptation using related signaling pathways[76,86]. Within a temperature range that does not significantly decrease photosynthetic efficiency, it is likely that for thermomorphogenesis as well the anabolic processes described here are relevant.

## Methods

### Plant material and growth conditions

We used the *Arabidopsis thaliana* genotypes (cv Columbia-0). *yuc2yuc5yuc8yuc9* was recrossed using all *yuc* alleles that are described in[18] except *yuc5-1* (SAIL_116_C0). We used the strain R-o-18 for *B. rapa* experiments. Oligonucleotides used for genotyping are listed in Supplementary Table 1. All materials used in the study are listed in Supplementary Table 2.

Seeds were size-selected and surface-sterilized using 70% (v/v) ethanol and 0.05% (v/v) Triton X-100 for 3 min followed by 10-min incubation in 100% (v/v) ethanol. Seeds were sowed on ½ Murashige and Skoog medium (½ MS) containing 0.8% (w/v) phytoagar and subsequently stratified at 4 °C for 3 days in darkness. For hypocotyl

elongation and RNA-seq experiments where seedlings were grown on vertical plates the phytoagar concentration was raised to 1.6% (w/v)[87]. For all experiments, seedlings were grown in 16 h/8 h, light/dark photoperiod (LD) at 21 °C in a Percival Scientific Model AR-22L (Perry, IA, USA) incubator. WL was emitted from white fluorescence tubes (Lumilux cool white 18 W/840) at a fluence rate of ~120 μmol m$^{-2}$ s$^{-1}$ and LRFR was achieved by supplementing WL with ~35 μmol m$^{-2}$ s$^{-1}$ FR light (LEDs λmax 740 nm) lowering the R (640–700 nm)/FR (700–760 nm) from 1.4 to 0.2, as measured by Ocean Optics USB2000 + spectrometer. A double layer of yellow filter (010 medium yellow, LEE Filters), lowering blue light from 27 μmol m$^{-2}$ s$^{-1}$ (WL) to 2.5 μmol m$^{-2}$ s$^{-1}$ (LB), was used to cover up the seedlings for LB treatments. The light spectra are shown in Supplementary Fig. 7.

For hypocotyl elongation and microscopy experiments, seedlings were grown for 4 days in WL (except Fig. 1a where seedlings were grown for 5 days) and subsequently kept in WL or transferred to light treatment (at ZT2) for additional 3 days. Fenpropimorph, picloram, sucrose, and sorbitol treatments were done on vertically grown seedlings on nylon meshes which were transferred to new plates containing the drug or the mock and put for 3 additional days into WL or light treatment (at ZT2). Fenpropimorph, picloram, and Concanamycin A were dissolved in DMSO (dimethylsulfoxide) applied at the indicated concentrations in Figure legends (DMSO for mock). For sucrose treatment (1%), the molar equivalent in sorbitol was used as an osmotic control.

For RNA-seq and western blot experiments, seedlings were grown for 5 days in WL and subsequently kept in WL or transferred to light treatment (at ZT2) for 3 h (RNA-seq) or 8 h (western blots) before harvesting. For concanamycin A treatment, seedlings were transferred to liquid ½ MS with shaking (75 rpm).

For ChIP-qPCR experiments, seedlings were grown in WL for 5 days and subsequently kept in WL or transferred to LRFR (at ZT2) for additional 5 days before harvesting.

For RT-qPCR, complex lipid and sterol measurement analysis, *B. rapa* seedlings were grown for 5 days in WL and subsequently kept in WL or transferred to LRFR (at ZT2) for the indicated time in legends before harvesting.

Seedlings imaging and measurement were done according to standard lab procedures and were described in detail[87].

## Constructs cloning

PCR amplifications were performed using Phusion® High-Fidelity DNA Polymerase. All cloning was done using In Fusion® HD Cloning kit. First, *GUSPlus:tOCS* was cloned in *pFP100* plasmid carrying *pAt2S3:GFP* selection marker[88] and the new plasmid was named as *pYI001*. *pFRO6* and *pGH3.17* were cloned into *pYI001* in order to obtain *pFRO6:GUSPlus:tOCS* and *pGH3.17:GUSPlus:tOCS*, respectively. *pUBQ10:SMT2-Flag:tOCS* was cloned into *pFP100*, while *pFRO6:SMT2-Flag*, and *pGH3.17:SMT2-Flag* were cloned into *pYI001*. The primers are listed in Supplementary Table 1. These constructs were transformed into *smt2-1* plants using *Agrobacterium tumefaciens* GV3101 strain by floral dip[89].

## RNA isolation, quantitative RT-PCR, and RNA sequencing

For RNA isolation, 5 days-old seedlings were harvested in liquid nitrogen and kept at −70 °C for overnight. Next day, seedlings were covered with −70 °C cold RNA*later*™-ICE and transferred to −20 °C overnight. Cotyledons and hypocotyls were dissected using sharp needles on top of an ice block under a binocular microscope (Nikon, SMZ1500) and RNA isolation and reverse transcription-quantitative polymerase chain reaction (RT-qPCR) reactions were performed as previously described[14]. In short, equal amounts of RNA were reverse transcribed into cDNA with Superscript II Reverse Transcriptase (Invitrogen). RT-qPCR was performed in three technical and three biological replicates (nine samples in total). Oligonucleotides are listed in Supplementary Table 1.

For RNA sequencing, RNA quality was assessed on a Fragment Analyzer (Agilent Technologies). From 40 ng total RNA, mRNA was isolated with the NEBNext Poly(A) mRNA Magnetic Isolation Module. RNA-seq libraries were then prepared from the mRNA using the NEBNext Ultra II Directional RNA Library Prep Kit for Illumina. Libraries were quantified by a fluorimetric method, and their quality assessed on a Fragment Analyzer (Agilent Technologies). Cluster generation was performed with the resulting libraries using Illumina HiSeq 3000/4000 SR Cluster Kit reagents. Libraries were sequenced on the Illumina HiSeq 4000 with HiSeq 3000/4000 SBS Kit reagents for 150 cycles. Sequencing data were demultiplexed with the bcl2fastq Conversion Software (v. 2.20, Illumina; San Diego, California, USA).

## ChIP-qPCR

10d-old seedlings *PIF4p:PIF4-HA* in *pif4-101*[49] and *PIF7p:PIF7-HA* in *pif7-2* seedlings[50] grown in WL for five days and then either kept in WL or transferred to LRFR for another five days were harvested in liquid nitrogen. Chromatin extraction was performed using standard procedures except that samples were cross-linked only with formaldehyde[90]. Immunoprecipitation was performed using an anti-HA antibody using standard procedures[91]. The qPCR was done in triplicates or quadruplicate on input and immunoprecipitated DNA. Peaks (P) were defined using a genome-wide ChIP study from etiolated seedlings that identified PIF4-peaks[92]. Controls (C) are on coding regions of each gene. Oligonucleotides are listed in Supplementary Table 1.

## Western blot analysis

Total protein extracts from seedlings were obtained as previously described[50]. In short, seedlings were ground in SDS-PAGE FSB (final sample buffer), heated at 95 °C for 5 min, centrifuges and the supernatant loaded on protein gels. Protein samples were separated on 10% Mini-Protean TGX gels and blotted on nitrocellulose membrane using Turbo transfer system. Membranes were blocked with 5% milk overnight at 4 °C or 1 h at room temperature for Anti-GFP JL-8 (1:4000), polyclonal H3 (1:2000), Anti-TUB (1:2000) antibodies before probing with horseradish peroxidase (HRP)-conjugated anti-rabbit (for H3) or anti-mouse (for anti-GFP and anti-TUB) as the secondary antibody (1:5000). Chemiluminescence signal were obtained with Immobilon Western Chemiluminescent HRP Substrate on an ImageQuant LAS 4000 mini (GE Healthcare, Buckinghamshire, UK). All western blot assays were done from four bioreps. Images were processed with ImageJ software.

## Microscopy and GUS staining

For DII-Venus microscopy and image quantification[14], seedlings were transferred to LRFR or kept in WL for 1 h before imaging. We used an inverted Zeiss confocal microscope (LSM 710, ×20 objective, 0.8 DIC). VENUS signal was detected using an Argon laser (excitation at 514 nm and bandpass emission between 520 and 560 nm). Image stacks (5–6/seedling) were acquired for every hypocotyl until the VENUS signal was lost. The pinhole was opened to collect the maximal signal intensity together with the minimal stack number (5.42 airy units, 20.2-μm section, 10.08-μm interval). We quantified the VENUS signal (ImageJ) via the SUM slices projection of four slices from the stack, excluding the first layer with the stomata.

For the lipid droplet (LD) quantification, seedlings were treated with BODIPY ™ 493/503 (2 μg/mL in H$_2$O) for 20 min at room temperature and imaged on inverted Zeiss confocal microscope (LSM 710, ×20 objective, 0.8 DIC) 30 h after the beginning of light treatment. BODIPY™ 493/503 signal was detected using an Argon laser (excitation at 488 nm and bandpass emission between 500 to 540 nm). Image stacks (6/seedling) were acquired for every hypocotyl. The pinhole was opened to 3.15 Airy Units (11.4 μm section, 10.00 μm interval). We quantified the signal (ImageJ) from a region of interest (ROI) on the

upper half of the hypocotyls and analyzed the fluorescence intensity of particles (size = 0–75 circularity = 0.50–1.00) in each image stack with a threshold (>3000) and summed the total intensity using the "Analyze particles" tool.

For chloroplast quantification, we used the same protocol as the LD quantification with the following changes. We used a HeNe633 laser (excitation at 633 nm and bandpass emission between 640 and 685 nm to have an equal contribution from ChlA and ChlB). The pinhole was opened to 3.15 Airy Units (11.4-µm section, 10.00-µm interval). The threshold was adjusted to >10,000; size and circularity were kept as default (0-infinity and 0–100, respectively) in ImageJ.

For visualization of autophagic bodies in *UBQ10:mCherry-ATG8e* lines[67,70], seedlings were transferred to LB or kept in WL for 8 h before imaging in the presence (5 µM) or absence (DMSO) of ConA. We used DPSS 561-10 laser (excitation at 561 nm and bandpass emission between 570 to 635 nm) (LSM 710, ×63 objective, 1.3 oil DIC). The pinhole was opened to 5.32 Airy Units (4.3-µm section, 4.21-µm interval). Two stacks are combined for each image.

The protocol for GUS staining reactions is described in ref. 50. In short, samples were fixed with ice-cold 90% acetone for 30 min, followed by washing twice with 50 mM $NaPO_4$ buffer (pH 7.2). Samples were incubated with staining solution (50 mM $NaPO_4$, 0.5 mM potassium ferricyanide, 0.5 mM potassium ferrocyanide, 0.1% triton X-100, and 2 mM X-gluc) overnight at 37 °C in the dark. De-staining was performed with ethanol series at room temperature until samples cleared. Cotyledons were prepared for cotyledon vasculature imaging as described in ref. 47. In short, seedlings were fixed in ethanol:acetic acid [3:1] and then rinsed in 70% and incubated in 100% ethanol at 4 °C overnight. For further clearing, seedlings were treated by 1 h of incubation in 10% NaOH at 42 °C and mounted in 50% glycerol. GUS staining and cotyledon vasculature were imaged using a dissecting microscope (Nikon SMZ1500).

## Sterol measurements

Four hypocotyls from 5 days-old *B. rapa* seedlings per sample were pooled and frozen in liquid nitrogen immediately after fresh weights were recorded. Samples were heated for 1 h in EtOH with 1% $H_2SO_4$ at 85 °C. Sterols were extracted in hexane. Free hydroxyl groups were derivatized at 110 °C for 30 min, surplus BSTFA-trimethylchlorosilane was evaporated, and samples were dissolved in hexane for analysis using GC-MS (Agilent 7890, A coupled to a mass spectrometer, MSD 5975, Agilent EI) under the conditions as described[93]. In short, An HP-5MS capillary column (5% phenyl-methyl-siloxane, 30-m, 250-mm, and 0.25-mm film thickness; Agilent) was used with helium carrier gas at 2 mL/min. The oven temperature was held at 200 °C for 1 min, then programmed with a 10 °C/min ramp to 305 °C (2.5-min hold) and a 15 °C/min ramp to 320 °C. Injection (1 µl) was done in splitless mode; injector and mass spectrometry detector temperatures were set to 250 °C. The ion source in a EI set at 70 eV, the mass range is 40–700 $m/z$ Raw data files were processed using the vendor-specific MassHunter (version B. 07.00, Agilent). Quantification of sterols was based on peak areas, which were derived from total ion current and using cholestanol as the internal standard. Each sterol was normalized to the total amount of detected sterols and presented as a percentage of the total.

## Sphingolipid measurements

Samples were collected as described in sterol measurements. For the analysis of sphingolipids by LC-MS/MS, lipids were extracted with Toledo solvent, dried, and then incubated 1 h at 50 °C in 2 mL of methylamine solution (7 mL methylamine 33% (w/v) in EtOH combined with 3 mL of methylamine 40% (w/v) in water (Sigma Aldrich) in order to remove phospholipids. After incubation, methylamine solutions dried at 40 °C under a stream of air. Finally, samples were resuspended into 100 µL of THF/MeOH/$H_2O$ (40:20:40, v/v) with 0.1% formic acid containing synthetic internal lipid standards (Cer d18:1/C17:0 and

GluCer d18:1/C12:0), thoroughly vortexed, incubated at 60 °C for 20 min, sonicated 5 min and transferred into LC vials.

LC-MS/MS (multiple reaction monitoring mode) analyses were performed with a model QTRAP 6500 (ABSciex) mass spectrometer coupled to a liquid chromatography system (1290 Infinity II, Agilent). Analyses were performed in the positive mode. Nitrogen was used for the curtain gas (set to 30), gas 1 (set to 30), and gas 2 (set to 10). Needle voltage was at +5500 V with needle heating at 400 °C; the declustering potential was adjusted between +10 and +40 V. The collision gas was also nitrogen; collision energy varied from +15 to +60 eV on a compound-dependent basis.

Reverse-phase separations were performed at 40 °C on a Super-colsil ABZ+, 100 × 2.1 mm column, and 5 µm particles (Supelco). The Eluent A was THF/ACN/5 mM Ammonium formate (3/2/5 v/v/v) with 0.1% formic acid, and eluent B was THF/ACN/5 mM Ammonium formate (7/2/1 v/v/v) with 0.1% formic acid. The gradient elution program for Cer and GluCer quantification was as follows: 0 to 1 min, 1% eluent B; 40 min, 80% eluent B; and 40 to 42, 80% eluent B. The gradient elution program for GIPC quantification was as follows: 0 to 1 min, 15% eluent B; 31 min, 45% eluent B; 47.5 min, 70% eluent B; and 47.5 to 49, 70% eluent B. The flow rate was set at 0.2 mL/min, and 5 mL sample volumes were injected.

The areas of LC peaks were determined using MultiQuant software (version 3.0; ABSciex) for sphingolipid quantification.

## Untargeted lipidomics mass spectrometry analysis

Four hypocotyls from 5 days-old *B. rapa* seedlings per sample were pooled, and preheated isopropanol (at 75 °C) was added immediately after fresh weights were recorded. Each sample with isopropanol was incubated at 75 °C for 15 min to inhibit phospholipase activity and cooled down to room temperature. Samples were kept at 4 °C overnight, and isopropanol was then evaporated to dryness using Nitrogen steam. Dry extracts were then reconstituted in 200 µL of IPA spiked with the internal standard mixture (SPLASH® LIPIDOMIX® Mass Spec Standard (92/8; v/v)). This solution was further homogenized in the Cryolys Precellys 24 sample Homogenizer (2 × 20 s at 10,000 rpm, Bertin Technologies, Rockville, MD, USA) with ceramic beads. The bead beater was air-cooled down at a flow rate of 110 L/min at 6 bar. Homogenized extracts were centrifuged for 15 min at 21,000×$g$ at 4 °C (Hermle, Gosheim, Germany) and the resulting supernatant was collected and transferred to an LC-MS vial.

Extracted samples were analyzed by reversed-phase liquid chromatography coupled to a high-resolution mass spectrometry (RPLC-HRMS) instrument (Agilent 6550 IonFunnel QTOF). In both, positive and negative ionization mode, the chromatographic separation was carried out on a Zorbax Eclipse Plus C18. Mobile phase was composed of A = 60:40 (v/v) Acetonitrile:water with 10 mM ammonium acetate and 0.1% acetic acid and B = 88:10:2 Isopropanol:acetonitrile:water with 10 mM ammonium acetate and 0.1% acetic acid. The linear gradient elution from 15 to 30% B was applied for 2 min, then from 30% to 48% B for 0.5 min, from 48 to 72% B, and the last gradient step from 72 to 99% B followed by 0.5 min isocratic conditions and a 3 min re-equilibration to the initial chromatographic conditions. The flow rate was 600 µL/min, column temperature 60 °C, and sample injection volume 2 µl.

ESI source conditions were set as follows: dry gas temperature 200 °C, nebulizer 35 psi and flow 14 L/min, sheath gas temperature 300 °C and flow 11 L/min, nozzle voltage 1000 V, and capillary voltage +/−3500 V. Full scan acquisition mode in the mass range of 100–1700 $m/z$ was applied for MS1 data acquisition while MS/MS data were acquired in the iterative data-dependent acquisition mode to facilitate lipid identification and annotation.

Pooled QC samples (representative of the entire sample set) were analyzed periodically (every six samples) throughout the overall analytical run in order to assess the quality of the data, correct the signal

intensity drift and remove the non-reproducible signals (CV >25%)[94]. In addition, a series of diluted quality controls (dQC) were prepared by dilution with isopropanol: 100% QC, 50% QC, 25% QC, 12.5% QC, and 6.25% QC, and analyzed at the beginning and at the end of the sample batch. This QC dilution series served as a linearity filter to remove the features that do not respond linearly (correlation with dilution factor is <0.65)[95].

**Data processing.** Raw LC-HRMS and HR(MS/MS) data were processed using MS-Dial software[96]. Relative quantification of lipids was based on EIC (Extracted Ion Chromatogram) areas for the monitored precursor ions at the MS1 level. Peak areas were normalized considering the sample amount (mg) (full lists of lipid species are given in Supplementary Data 9). The obtained data (containing peak areas of detected and identified lipids by MS and MS/MS, and using MS only across all samples) were exported to "R" software http://cran.r-project.org/ where the signal intensity drift correction was done within the LOWESS/Spline normalization program[96] followed by noise filtering (CV (QC features) >30%) and visual inspection of linear response.

The abundance of each MS/MS-detected lipid species was normalized to the total amount of MS/MS-detected lipids and presented as a percentage of the total.

### RNA-seq initial data analysis
Sequence reads from FASTQ files were processed as follows: Reads that did not pass Illumina's filtering were removed. Adapters and low-quality 3′ ends were trimmed from the reads using Cutadapt (v1.8)[97]. Reads matching to ribosomal RNA sequences were removed with fastq_screen (v. 0.11.1). Remaining reads were further filtered for low complexity with reaper (v. 15-065)[98]. More than 30 million uniquely mapped reads were obtained per library, and reads were aligned against the *Arabidopsis thaliana*.TAIR10.39 genome using STAR (v. 2.5.3a)[99]. The number of read counts per gene locus was summarized with htseq-count (v. 0.9.1)[100] using the *Arabidopsis thaliana*.TAIR10.39 gene annotation. The quality of the RNA-seq data alignment was assessed using RSeQC (v. 2.3.7)[101].

Statistical analysis was performed for genes independently in R (R version 4.0.2). All steps described here were performed separately for the samples from hypocotyls and cotyledons (except for the initial clustering of all samples together). Genes with low counts were filtered out according to the rule of 1 count(s) per million (cpm) in at least one sample. The number of genes retained in the analyses based on this filtering is different for hypocotyls and cotyledons. Library sizes were scaled using TMM normalization. Subsequently, the normalized counts were transformed to cpm values and a log2 transformation was applied by means of the function cpm with the parameter setting prior.counts = 1 (edgeR)[102].

Differential expression was computed with the R Bioconductor package "limma"[103] by fitting data to a linear model. The approach limma-trend was used. Fold changes were computed and a moderated *t* test was applied. *P* values were adjusted using the Benjamini–Hochberg (BH) method, which controls for the false discovery rate (FDR), globally across several comparisons of experimental conditions. The adjustment was performed in a few different ways customized to different parts of the data analysis. For Supplementary Data 1, Fig. 1, and Supplementary Fig. 1, *P* values were adjusted globally across pairs of comparisons, LB vs. WL and LRFR vs. WL in each genotype separately. For Supplementary Data 5, Fig. 3, and Supplementary Fig. 3, *P* values were adjusted globally across three comparisons between different genotypes while keeping the light condition (WL) unchanged: *pif457* vs Col-0, *yuc2589* vs Col-0 and *smt2-1* vs Col-0. For Supplementary Data 3 and 4; Figs. 2 and 3; and Supplementary Fig. 2 and 3, *F* tests were used that yielded a single *P* value for three comparisons and post hoc testing as

implemented; the R package limma was then applied to identify significantly regulated genes per comparison. For Supplementary Fig. 5, the same *F* test is applied for two comparisons.

### Gene set enrichment analysis for gene ontology
Gene set enrichment analyses were conducted with ShinyGO v0.61:Gene Ontology Enrichment Analysis + more (http://bioinformatics.sdstate.edu/go/)[104] in *Arabidopsis thaliana* using a *P* value cutoff (FDR) of 0.05 and 500 most significant terms to show. The networks of enriched GO categories were visualized with R software (https://www.r-project.org/) using "visNetwork" and "igraph" libraries. Two terms (nodes) were connected if they share 20% or more genes. The size of the nodes indicates fold change for each term. We highlighted selected terms for each organ and light condition that we could easily relate to growth regulation (Fig. 1d and Supplementary Fig. 1c). The highlighted terms are not necessarily the most significant ones (full lists are available in Supplementary Data 2 and as interactive versions).

### Statistical motif analysis in promoter or upstream gene sequences
Motif enrichment analyses were conducted with TAIR's Motif Analysis tool (https://www.arabidopsis.org/tools/bulk/motiffinder/index.jsp) using 1-kb upstream sequences.

### Other statistical analyses and data representation
For all the phenotypic analyses of hypocotyl elongation and the quantification of DII-signal, we performed two-way analysis of variance (ANOVA) (aov) and computed Tukey's Honest Significance Differences (HSD) test ("agricolae" package) with default parameters using R software. For phenotypic analysis of treatments with fenpropimorph and picloram, we performed two-way analysis of variance (ANOVA) (aov) and represented the significance as genotype*drug interaction in the given light conditions. For the comparisons including qPCR, ChIP-qPCR, sterol measurements, lipidomics analysis and autophagic flux assays, we performed Student's *T* test. For lipidomics analysis, we further applied a BH correction for *P* values. We used binomial distribution for the PIF4 target enrichment, promoter motif, and to determine the significance of PIFs and/or YUCs dependence of enriched GO terms in given light conditions (Supplementary Data 3 and Supplementary Figs. 4 and 5).

In boxplots, the horizontal bar represents the median; boxes extend from the 25th to the 75th percentile, whiskers extend to show the data range. In data point plots, points are given as individual values from biological replicates, the horizontal lines indicate median, and error bars extend to show data range. For the fenpropimorph and picloram treatments, the data show the means ± SD with a simple linear regression line connecting data points for the given light condition and the genotype. Asterisks (*) and different letters in the graphs indicate significant differences as defined by the statistical methods described above (*P* < 0.05).

### Reporting summary
Further information on research design is available in the Nature Research Reporting Summary linked to this article.

## Data availability
The RNA-seq data discussed in this publication have been deposited in NCBI's Gene Expression Omnibus[105] and are accessible through GEO Series accession number GSE174655. The MS data for lipid measurements were deposited in Metabolights (https://www.ebi.ac.uk/metabolights/). Untargeted lipidomic analysis with the unique identifier MTBLS2796, MTBLS5753 for the sphingolipidomics, and MTBLS5766 for sterols. Source data are provided with this paper.

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

## Acknowledgements

We thank Martina Legris, Laure Allenbach Petrolati, Olivier Michaud, Mieke de Wit, Anupama Goyal, Ana Lopez Vazquez, Ganesh Mahadeo Nawkar, and Maud Lagier for providing resources, technical support and/or comments on the manuscript; René Dreos for advice on statistical methods; Niko Geldner, Ersin Gül (ETH Zurich), Sinem Celebioven (University Zurich), Soner Yildiz (Icahn School of Medicine) and Yasin Dagdas (GMI, Vienna) for suggestions and comments on the manuscript; Christian Hardtke, Teva Vernoux (ENS Lyon), Richard Vierstra (Washington University, St Louis), Dany Geelen (University Gent) and Yasin Dagdas for providing resources. We are grateful to the CIF (https://cif.unil.ch/) for help with microscopy, the GTF (https://wp.unil.ch/gtf/) for RNA-seq. experiments and the Metabolomics Unit (https://www.unil.ch/metabolomics/en/home.html) for the lipidomic analysis and the Bordeaux-Metabolome platform for sterol analysis (https://www.biomemb.cnrs.fr/en/lipidomic-plateform/). Work in the Fankhauser lab is supported by the University of Lausanne and the Swiss National Science Foundation (310030B_179558) and the Velux Foundation (Project 1455). The Bordeaux Metabolome-Lipidome Facility-MetaboHUB by a grant from ANR (no. ANR–11–INBS–0010). Manon Genva thanks the National Fund for Scientific Research (FNRS, Belgium, through grant 2022/V 3/5/053-40010130-JG/JN-2724) and the University of Liège for mobility grants.

## Author contributions

Y.C.I.: conceptualization, formal analysis, data curation, methodology, investigation, resources, validation, visualization, and writing—original draft. J.K.: conceptualization, investigation, formal analysis, and validation. A.S.F., M.T., and V.C.G.: investigation, resources, and validation. S.P. and L.W.: formal analysis, data curation, and methodology. S.M.: conceptualization, data curation, validation, investigation, and methodology. L.F., M.G., and P.V.D.: investigation. J.I and H.G.A.: investigation, data curation, and validation. C.F.: conceptualization, data curation, methodology, project administration, resources, funding acquisition, supervision, validation, and writing—original draft. All authors read and approved the manuscript.

## Competing interests

The authors declare no competing interests.
