## [Peer Review File · Nature Communications]

A combination of plasma membrane sterol biosynthesis and autophagy is required for shade-induced hypocotyl elongationReviewer #1 (Remarks to the Author):

The corresponding author's group previously reported that cell elongation-related genes, such as those involved in cell wall remodeling and sterol biosynthesis, were up-regulated in hypocotyls exposed to a low red to far-red light condition (LRFR) (Kohnen et al., 2016). In this manuscript, Ince et al. extended their RNA-Seq-based approach and additionally interrogated the transcriptome responses of hypocotyls and cotyledons under low blue light condition (LB), because the combination of LB and LRFR better represents vegetative shade than LRFR alone. They found that PIFs specifically induced the expression of sterol biosynthetic genes in hypocotyls exposed to LRFR. In LB-treated hypocotyls, starvation symptom was identified and autophagic flux was induced. Thus, the authors proposed that vegetative shade enhances hypocotyl growth by combining autophagy-mediated recycling and promotion of specific lipid biosynthetic processes (lines 34-36).

1. The authors need to clarify their proposition and refine the data that could explain how LB-induced autophagy relates to either light signaling components or nutrition status. If PIFs and YUC genes are largely not essential for transcriptional regulation of ATG genes and other starvation response genes (as indicated by Figure 3 and conclusion in lines 184 and 185), then autophagy induction is more likely due to fixed carbon starvation, which results from a significant reduction in PAR. Both *atg7* and *atg5* mutants had a reduced hypocotyl elongation in all four conditions, compared with Col-0 (Figure 7e and Supplementary Figure 7g). This phenotype supports the notion that (fixed carbon and other nutrients recycled from) autophagy is generally necessary for seedling growth and hypocotyl elongation, rather than "autophagy is particularly important to promote hypocotyl elongation in LB" (lines 345-346). A few control experiments can be done to test these possibilities. On the one hand, seedlings can be supplemented with exogenous sucrose to prevent carbon limitation. On the other hand, red light may be filtered out to reduce PAR similarly by ~70%. Autophagy assay and hypocotyl elongation test using Col-0, *smt2*, *atg7*, and *smt2 atg7* seedlings in these conditions would help determine whether a specific light condition or nutrient status is important in this biological context. More comprehensive autophagy assays of dissected samples – cotyledons, hypocotyls, and roots – are also recommended, given the notion that autophagy is regulated spatially and temporally.

2. The fluence spectrum of LB+LRFR can be added to Supplementary Figure 7a. This will help interpreting data shown in Figure 7, by excluding the possibility of any confounding effect by both blue light filtering and FR light addition.

3. No quantification in autophagy assays is provided in Figure 7b and 7f.

4. How is the difference in the sterol lipid content between WL and LRFR samples (Figure 6a; at 30 h)? The difference appears statistically significant, but there is no mentioning about this difference in the main text.

5. The authors described "in vegetative shade the combination of sterol biosynthesis and autophagy is essential for hypocotyl growth promotion" (lines 33-34). Experiments using plants grown under vegetative canopy (e.g., Figure 2 of de Wit et al., 2016) will reinforce the findings of this work.

6. There are statements that need rephrasing or clarification:

Lines 178-179: "As in LRFR, most WL-misregulated genes in hypocotyls required both PIFs and YUCs". This sounds like circular reasoning.

Lines 1088: "hypocotyl-induced genes in LB". Should it be "LB-induced genes in hypocotyls"?

Reviewer #2 (Remarks to the Author):

The present study reveals the requirement of a combination of plasma membrane sterol biosynthesis and autophagy for shade-induced *Arabidopsis* hypocotyl elongation. First of all, the authors performed RNA sequencing and bioinformatics assays to explore the effects of low blue light (LB) and a low red to far-red ratio (LRFR) on gene expression, and found that LB and LRFR induced distinct transcriptional changes in elongating hypocotyls. Specifically, they showed a significant difference between the treatments with LB upregulating numerous catabolic processes while LRFR inducing many anabolic processes. They then performed GO assay and found that LRFR-induced genes in the hypocotyls largely depends on auxin transported from the cotyledons with a potential local action of PIFs in hypocotyls, whereas the majority of LB-induced genes do not require PIFs or YUCs. Through ChIP-qPCR, sterol measurements and untargeted lipidomics mass spectrometry analyses, the authors showed that PIFs-induced SMT2 expression in the hypocotyls promotes growth in LRFR and that LRFR selectively promotes the accumulation of PM lipids. Moreover, the authors found that autophagy is important to promote hypocotyl elongation under LB conditions. This work shows that, under vegetational shade comprising LB and LRFR, hypocotyl elongation requires autophagy and the induction of specific anabolic processes, which is of significant novelty in the field. However, the authors need to address the comments and concerns I raise below.

Major points:

1. No seedlings' pictures showing the hypocotyl phenotypes are available in this paper, and the error bars for the statistic assays of the hypocotyl phenotypes were too large (Figures 5c and 7e). In addition, the number of samples for statistics analysis is not enough, which should be more than or equal to 30.
2. In the present study, sterol measurement and untargeted lipidomics mass spectrometry analysis were performed in *B. rapa* but not in *Arabidopsis thaliana*. Since there may be some major differences in these processes among different species, these data obtained from *B. rapa* may not be convincing to explain the biological processes in *Arabidopsis thaliana*.
3. The authors chose ATG8e to observe autophagic bodies, while chose ATG8a to detect free GFP accumulation. As ATG8 is found in the autophagosomal membranes, it is suggested that the authors use both ATG8a and ATG8e to visualize the autophagic bodies and detect the level of free GFP.
4. The authors stated that LB induced the degradation of GFP-ATG8, which is not consistent with the data showing that the amount of free GFP in WL is much more than that under LB (Fig. S7d). Moreover, although the authors stated that LB induces autophagy, there was basically no difference in the number of autophagosomes under WL and LB (Fig. S7c). Given the authors' demonstration that LB induced the degradation of GFP-ATG8, why is there no difference in GFP-ATG8a protein levels in WT and autophagy mutant *atg7* under LB (Fig. S7e)?

Minor points:

1. In Figure 4d, why are *smt2* hypocotyls shorter than WT, while *smt3* hypocotyls taller than WT in LB?
2. In Figure 6b, why were the middle and upper parts of the hypocotyls used to analyze? In Figure 7b, why were the cotyledons used to detect, but not the hypocotyls or roots?
3. In Fig. S7e, the sample loading was not equal.
4. The model diagram needs to be revised properly (Figure 8), because it is not fully supported by the experimental results obtained in this study.

Reviewer #3 (Remarks to the Author):

This is a generally well written and documented paper focusing on mechanisms of shade-induced hypocotyl elongation, which I enjoyed reading. An elegant combination of transcriptomics and mutants exposed to different light conditions is used to generate extensive datasets to dissect the different known or suspected pathways involved in this response. As far as I can the global transcript analysis is expertly done, and I have no specific concerns regarding this aspect of the paper.

At some point the work begins to focus on sterol biosynthesis as key genes of this pathway are regulated in specific ways under certain treatments. A hypothesis is tested postulating that sterols are found primarily in the plasma membrane, which needs to expand as hypocotyl cells elongate, and based on previous observations that fixed carbon is directed into lipid biosynthesis under these same conditions. The data could generally support this hypothesis, but I do have a few points that should be considered:

- 1. I do not find the pharmacological results using a sterol biosynthesis inhibitor as shown in Figure 4e very convincing because the errors are large, only two concentrations were used, and the effects are pretty minor.**
- 2. The interpretation of the lipidomics data as shown in Figure 6 may not necessarily support the conclusions. Effects are fairly small, and what is more concerning is that there is no effect on sphingolipids under LRFR, although these are also found primarily in plasma membranes, perhaps more so than sterols.**
- 3. Glycerophospholipids are not exclusive or predominant in plasma membranes as chloroplast certainly contain PG and PA and have more bulk than plasma membranes. Hence the interpretation of the data as described may be questionably.**
- 4. I am wondering whether diversion of fixed carbon into lipids under certain conditions is really necessary to support the relatively small amount of carbon ending up in plasma membranes due to elongation.**

In summary, I have a hard time to become convinced that the importance of sterol biosynthesis under certain conditions is due to the need for plasma membrane expansion when hypocotyls elongate. I could easily imagine that sterol derived signal compounds other than brassinosteroids are important under these conditions. I have a have made a few minor edits in the attached PDF.

Response to the reviewers' comments (NCOMMS-21-40466)

Our answers are in **bold**

We would like to take this opportunity to thank the reviewers for their constructive comments that we first address by pointing out the major modifications in the revised manuscript and then point by point.

General comments about the revised manuscript

To address the reviewers comments our main new findings are presented in the following figures of the revised manuscript.

- **Figures 8 a-c, S9 b, c and d. Comparing the effects of low blue (LB) versus low PAR (LP) on autophagy and hypocotyl elongation. This allowed us to conclude that it is low blue and not low PAR (PAR is reduced in the LB treatment) that induces autophagy and hypocotyl elongation.**
- **Figure 7 e and f, quantification of LB-induced autophagy in hypocotyls and cotyledons**
- **Figure S6c, a dedicated analysis of Sphingolipids (major plasma membrane lipids)**
- **Figure S7, spectra of all light treatments.**
-

This and modifications to improve clarity leads to new figures 6 to 8 and S6 to S9.

Our quantification of autophagy in hypocotyls and cotyledons (Fig. 7e, f) was done using a biochemical rather than a microscopic assay. The reason for this choice is because we discovered that tissue penetration of ConA, a drug inhibiting vacuolar ATPases, is not optimal in aerial tissues. ConA must penetrate well in order to see autophagosomes microscopically. We found that pretty much all studies in Arabidopsis using the microscopic assay are done in roots, a tissue in which most drugs penetrate much more easily than in aerial parts. We discussed this problem at length with Dr Yasin Dagdas (GMI, Vienna), an autophagy specialist, who confirmed the difficulty of microscopic examination of autophagy in aerial tissues. Specific answers are provided below

Reviewer #1 (Remarks to the Author):

The corresponding author's group previously reported that cell elongation-related genes, such as those involved in cell wall remodeling and sterol biosynthesis, were up-regulated in hypocotyls exposed to a low red to far-red light condition (LRFR) (Kohnen et al., 2016). In this manuscript, Ince et al. extended their RNA-Seq-based approach and additionally interrogated the transcriptome responses of hypocotyls and cotyledons under low blue light condition (LB), because the combination of LB and LRFR better represents vegetative shade than LRFR alone. They found that PIFs specifically induced the expression of sterol biosynthetic genes in hypocotyls exposed to LRFR. In LB-treated hypocotyls, starvation symptom was identified and autophagic flux was induced. Thus, the authors proposed that vegetative shade enhances hypocotyl growth by combining autophagy-mediated recycling and promotion of specific lipid biosynthetic processes (lines 34-36).

1. The authors need to clarify their proposition and refine the data that could explain how LB-induced autophagy relates to either light signaling components or nutrition status. If PIFs and YUC genes are largely not essential for transcriptional regulation of ATG genes and other starvation response genes (as indicated by Figure 3 and conclusion in lines 184 and 185), then autophagy induction is more likely due to fixed carbon starvation, which results from a significant reduction in

PAR. Both *atg7* and *atg5* mutants had a reduced hypocotyl elongation in all four conditions, compared with Col-0 (Figure 7e and Supplementary Figure 7g). This phenotype supports the notion that (fixed carbon and other nutrients recycled from) autophagy is generally necessary for seedling growth and hypocotyl elongation, rather than “autophagy is particularly important to promote hypocotyl elongation in LB” (lines 345-346). A few control experiments can be done to test these possibilities. On the one hand, seedlings can be supplemented with exogenous sucrose to prevent carbon limitation. On the other hand, red light may be filtered out to reduce PAR similarly by ~70%. Autophagy assay and hypocotyl elongation test using Col-0, *smt2*, *atg7*, and *smt2 atg7* seedlings in these conditions would help determine whether a specific light condition or nutrient status is important in this biological context. More comprehensive autophagy assays of dissected samples – cotyledons, hypocotyls, and roots – are also recommended, given the notion that autophagy is regulated spatially and temporally.

Answer. New experiments addressing this important question are presented on Figures 7e, 7f (quantification of LB-induced autophagy in cotyledons and hypocotyls), Figures 8 a-c, S9 b, c and d (light feature inducing autophagy and hypocotyl elongation). Collectively these experiments allowed us to conclude that LB induces autophagy in hypocotyls and cotyledons. We could also show that LB and not an equivalent decrease in PAR (66% of our white light condition) induces autophagy (Figure 8a, b). Moreover, while LB induces hypocotyl elongation a 66% decrease in PAR (keeping the same light spectrum) did not. In addition, performing a LB treatment with PAR corresponding to PAR in white light also leads to longer hypocotyls (Figures 8c, S9b, c). Finally, sucrose (compared to sorbitol) leads to a substantial increase in hypocotyl elongation in *atg7* but also in the WT (as reported previously) (Figure S9d). We did not analyze autophagy in roots for the following reasons (i) the low R/FR-induced growth response of roots is totally different from the one in hypocotyls (e.g. van Gelderen et al., 2018) (ii) in response to low R/FR more fixed carbon is allocated to hypocotyls but not to roots (de Wit et al., 2018) (iii) our manuscript focusses on hypocotyl growth responses not what is happening in roots.

2. The fluence spectrum of LB+LRFR can be added to Supplementary Figure 7a. This will help interpreting data shown in Figure 7, by excluding the possibility of any confounding effect by both blue light filtering and FR light addition.

Answer. The spectra of all our light treatments is presented in the new version of figure S7

3. No quantification in autophagy assays is provided in Figure 7b and 7f.

Answer. (1) As explained in the general comments (page 1), we discovered that microscopic assays to analyze autophagy in aerial parts are particularly challenging. We discussed this at length with Dr Yasin Dagdas (GMI, Vienna), an autophagy specialist, who confirmed the difficulty of microscopic examination of autophagy in aerial tissues. The microscopic data shown on Figures 7d, S8d and S8e, validate our assays (autophagosomes not observed without ConA or in *atg5-1*). This being said, we believe that this assay is rather of qualitative nature (LB vs. WL in 7d). Nevertheless, images have been analyzed in double blind by multiple lab members who counted significantly more autophagomes in LB than WL samples (I can provide you with this data). (2) We provide quantitative analyses of the western blot assay on Figures 7c (autophagic flux), 7f (autophagy in hypocotyls and cotyledons) and 8b (autophagy in LB vs low PAR).

4. How is the difference in the sterol lipid content between WL and LRFR samples (Figure 6a; at 30 h)? The difference appears statistically significant, but there is no mentioning about this difference in the main text.

Answer. The non-targeted lipidomic analysis presented in figure 6a is poorly suited for the analysis of sterols and sphingolipids (2 major classes of PM lipids). For this reason, we performed dedicated analyses for these two lipid classes (different extraction procedure & analyses), data are shown on figures S6b and c. This dedicated analysis is much more reliable for these lipid classes, which is why we now only show the data for these specific analyses. Based on this data we conclude that a low R/FR treatment does not lead to a significant change in the composition of these lipids (figures S6b and c).

5. The authors described “in vegetative shade the combination of sterol biosynthesis and autophagy is essential for hypocotyl growth promotion” (lines 33-34). Experiments using plants grown under vegetative canopy (e.g., Figure 2 of de Wit et al., 2016) will reinforce the findings of this work.

Answer. The concept that the combination of LB and LRFR is a good mimic of vegetational shade is well established (e.g. de Wit et al., 2016). We have therefore not tried to obtain a real canopy from other plants and grown seedlings underneath, as it is hard to do this in a controlled manner (obtaining a reproducible canopy cover).

6. There are statements that need rephrasing or clarification:

Lines 178-179: “As in LRFR, most WL-misregulated genes in hypocotyls required both PIFs and YUCs”. This sounds like circular reasoning.

Answer: we modified the text to improve clarity

Lines 1088: “hypocotyl-induced genes in LB”. Should it be “LB-induced genes in hypocotyls”?

Answer: we modified the text to improve clarity

Reviewer #2 (Remarks to the Author):

The present study reveals the requirement of a combination of plasma membrane sterol biosynthesis and autophagy for shade-induced Arabidopsis hypocotyl elongation. First of all, the authors performed RNA sequencing and bioinformatics assays to explore the effects of low blue light (LB) and a low red to far-red ratio (LRFR) on gene expression, and found that LB and LRFR induced distinct transcriptional changes in elongating hypocotyls. Specifically, they showed a significant difference between the treatments with LB upregulating numerous catabolic processes while LRFR inducing many anabolic processes. They then performed GO assay and found that LRFR-induced genes in the hypocotyls largely depends on auxin transported from the cotyledons with a potential local action of PIFs in hypocotyls, whereas the majority of LB-induced genes do not require PIFs or YUCs. Through ChIP-qPCR, sterol measurements and untargeted lipidomics mass spectrometry analyses, the authors showed that

PIFs-induced SMT2 expression in the hypocotyls promotes growth in LRFR and that LRFR selectively promotes the accumulation of PM lipids. Moreover, the authors found that autophagy is important to promote hypocotyl elongation under LB conditions. This work shows that, under vegetational shade comprising LB and LRFR, hypocotyl elongation requires autophagy and the induction of specific anabolic processes, which is of significant novelty in the field. However, the authors need to address the comments and concerns I raise below.

Major points:

1. No seedlings' pictures showing the hypocotyl phenotypes are available in this paper, and the error

bars for the statistic assays of the hypocotyl phenotypes were too large (Figures 5c and 7e). In addition, the number of samples for statistics analysis is not enough, which should be more than or equal to 30.

Answer. We now present representative pictures of seedlings on Figure S9b. There is no particular reason why one should show data from more than 30 seedlings for each experiment. This being said, this is what we do for the vast majority of experiments (indicated in the figure legends).

2. In the present study, sterol measurement and untargeted lipidomics mass spectrometry analysis were performed in *B. rapa* but not in *Arabidopsis thaliana*. Since there may be some major differences in these processes among different species, these data obtained from *B. rapa* may not be convincing to explain the biological processes in *Arabidopsis thaliana*.

Answer. We understand the reviewers concern but would like to point out that *Brassica rapa* and *Arabidopsis thaliana* are relatively closely related species. We explain why some biochemical experiments are impossible in *Arabidopsis* forcing us to use *Brassica*. We believe that conclusions from both species rather reinforce our conclusions. Our cell biology experiment in *Arabidopsis* presented in figures 6b and 6c confirm the biochemical analyses performed in *Brassica* (Figure 6a). In low R/FR storage lipids and chloroplast lipids decline in both species. The regulation of *SMT2/3* expression in *Brassica* and *Arabidopsis* is similar (Figure 2 and S6).

3. The authors chose ATG8e to observe autophagic bodies, while chose ATG8a to detect free GFP accumulation. As ATG8 is found in the autophagosomal membranes, it is suggested that the authors use both ATG8a and ATG8e to visualize the autophagic bodies and detect the level of free GFP.

Answer. Using different tools to study autophagy (GFP-ATG8a and RFP-ATG8e) and different approaches (microscopy and cleavage assay revealed by western blotting) provides more evidence for the fact that in LB there is more autophagy. Based on our own observations and discussion with Yasin Dagdas we found that RFP-ATG8e works better for microscopic examination of autophagy explaining our approach.

4. The authors stated that LB induced the degradation of GFP-ATG8, which is not consistent with the data showing that the amount of free GFP in WL is much more than that under LB (Fig. S7d). Moreover, although the authors stated that LB induces autophagy, there was basically no difference in the number of autophagosomes under WL and LB (Fig. S7c). Given the authors' demonstration that LB induced the degradation of GFP-ATG8, why is there no difference in GFP-ATG8a protein levels in WT and autophagy mutant *atg7* under LB (Fig. S7e)?

Answer. There appears to be some misunderstanding here, which I will try to clarify. Former figure S7d (now S8b) shows that LB (compared to white light) does not change the accumulation of free GFP in a plant expressing GFP (not GFP-ATG8) from a constitutive promoter. This is a control indicating that GFP accumulation itself is not altered by the LB treatment. As pointed out by the reviewer there is no difference between the number of autophagosomes between LB and white light in former figure S7c (now S8d). This is because in this experiment seedlings were not treated with ConA, therefore it is not possible to see the accumulation of autophagosomes. Moreover, in LB + ConA in *atg5-1* we also do not see autophagosomes (Figure S8e). Finally, regarding former figure S7e (now S8c), in LB we see the appearance of free GFP in the WT but not in the *atg7* mutant, which shows genetically this LB induced cleavage of GFP-ATG8a depends on autophagy.

Minor points:

1. In Figure 4d, why are *smt2* hypocotyls shorter than WT, while *smt3* hypocotyls taller than WT in LB?

Answer. In figure 4d *smt2-1*, *smt2-2* and *smt3-1* are not statistically significantly different from the WT (Col-0). However, the reviewer is correct in noting that there is a tendency for *smt2* alleles to be a bit shorter than the WT in LB. This could be because SMT2 also plays a role in LB as indicated by the finding that in an *atg7smt2* double mutant hypocotyl growth is strongly affected in LB (Figure 8e).

2. In Figure 6b, why were the middle and upper parts of the hypocotyls used to analyze? In Figure 7b, why were the cotyledons used to detect, but not the hypocotyls or roots?

We analyzed the middle and upper part of hypocotyls in figure 6 because there are more chloroplasts in the upper than lower hypocotyl. For figure 7b we used the microscopic assay in cotyledons because it appears to work better in cotyledons than hypocotyls. As discussed in the general comments it appears that penetration of ConA into aerial parts is not optimal rendering this assay challenging. It is possible that ConA penetrates more easily into cotyledons than hypocotyls. Therefore, we now present the western blot GFP-ATG8a cleavage assay to show that autophagy occurs both in hypocotyl and cotyledons (Figure 7e, f). Finally, we did not analyze roots because the growth response and resource reallocation during shade responses is totally different in hypocotyls and roots. See also answer to reviewer 1.

3. In Fig. S7e, the sample loading was not equal.

Answer: Former Figure S7e is now S8c. The loading (TUB) is quite similar, if anything there is more loading in *atg7* which does not show any free GFP. It therefore does not alter the conclusion from this experiment.

4. The model diagram needs to be revised properly (Figure 8), because it is not fully supported by the experimental results obtained in this study.

Answer: Our new experiments show that it is LB rather than low PAR that induces autophagy. The model was revised accordingly.

Reviewer #3 (Remarks to the Author):

This is a generally well written and documented paper focusing on mechanisms of shade-induced hypocotyl elongation, which I enjoyed reading. An elegant combination of transcriptomics and mutants exposed to different light conditions is used to generate extensive datasets to dissect the different known or suspected pathways involved in this response. As far as I can the global transcript analysis is expertly done, and I have no specific concerns regarding this aspect of the paper.

At some point the work begins to focus on sterol biosynthesis as key genes of this pathway are regulated in specific ways under certain treatments. A hypothesis is tested postulating that sterols are found primarily in the plasma membrane, which needs to expand as hypocotyl cells elongate, and based on previous observations that fixed carbon is directed into lipid biosynthesis under these same conditions. The data could generally support this hypothesis, but I do have a few points that should be considered:

1. I do not find the pharmacological results using a sterol biosynthesis inhibitor as shown in Figure 4e

very convincing because the errors are large, only two concentrations were used, and the effects are pretty minor.

Answer: We understand the concern of the reviewer however, the data presented on Figure 4e is confirmed by the data shown on Figure S4d in which we used more sterol biosynthesis inhibitor concentrations.

2. The interpretation of the lipidomics data as shown in Figure 6 may not necessarily support the conclusions. Effects are fairly small, and what is more concerning is that there is no effect on sphingolipids under LRFR, although these are also found primarily in plasma membranes, perhaps more so than sterols.

Answer: This is an important comment and we realized that our untargeted lipidomics analysis is poorly suited for the analysis of major plasma membrane lipids: sterols and sphingolipids. We already conducted dedicated an analysis of sterols and have now also included a dedicated analysis of sphingolipids shown on new figure S6c. Moreover, we removed the data about sterols and sphingolipids from the untargeted analysis (modified figure 6a). Our data using dedicated approaches to study sterols and sphingolipids indicate that there is no major change in composition within those lipid classes. Taken together with the overall up-regulation in response to low R/FR of genes involved in sterol and sphingolipid biosynthesis our data suggests that transcriptional regulation underlies the need for more overall synthesis of those lipid classes.

3. Glycerophospholipids are not exclusive or predominant in plasma membranes as chloroplast certainly contain PG and PA and have more bulk than plasma membranes. Hence the interpretation of the data as described may be questionably.

Answer: Our transcriptomic data shows that terms like sterol biosynthesis and sphingolipid biosynthesis are upregulated in the hypocotyl of low R/FR treated seedlings; *these two classes of lipids being greatly enriched in the plasma membrane*. Our targeted and untargeted lipidomic analysis (in Brassica) shows that typical storage and chloroplast lipids (*PG and PA are minor lipids and glyco-glycero-lipids MGDG DGDG are major lipids of chloroplast membranes, called GDG in Figure 6*) decline in response to low R/FR. The same is observed in Arabidopsis seedlings using cell biological approaches (Figure 6). Hence, *as stated by the reviewer*, while glycerophospholipids are not exclusively found in plasma-membrane, we feel that the increase in *sterol and sphingolipid* in response to low R/FR is consistent with the idea of enhanced synthesis of plasma membrane lipids.

4. I am wondering whether diversion of fixed carbon into lipids under certain conditions is really necessary to support the relatively small amount of carbon ending up in plasma membranes due to elongation.

Answer : I am not sure what allows the reviewer to conclude that a small amount of carbon ends up in plasma-membranes in conditions promoting hypocotyl elongation. Based on our pulse label experiment the amount of freshly fixed carbon being reallocated to lipids in the hypocotyls is 2-3X higher in shade compared to the sun (de Wit et al., 2018). Moreover, unlike primary cell walls which are flexible, lipid bilayers are not hence if a cell gets longer (as is the case in hypocotyls of low R/FR treated seedlings) the plasma-membrane and vacuole have to grow accordingly.

In summary, I have a hard time to become convinced that the importance of sterol biosynthesis under certain conditions is due to the need for plasma membrane expansion when hypocotyls elongate. I could easily imagine that sterol derived signal compounds other than brassinosteroids are

important under these conditions.

I have a have made a few minor edits in the attached PDF.

Answer : increased signaling sterol production in low R/FR may also contribute to the growth response. However, more plasma-membrane lipids are required to sustain cell elongation and collectively our data is fully consistent with this being an important element.

We thank the reviewer for his edits that we corrected.

Reviewer #1 (Remarks to the Author):

The revised manuscript by Ince et al. is significantly improved and they sufficiently addressed all of my concerns. However, I am with reviewer 3 in that the lipid analysis data do not fully support their conclusion. I think that the authors should tone down and rephrase their description and conclusions about lipid analysis. For example, instead of saying "LRFR promotes anabolism including biosynthesis of plasma-membrane sterols downstream of PHYTOCHROME-INTERACTING FACTORS (PIFs) acting in hypocotyls (lines 33-35)", they could describe "LRFR specifically induced expression of sterol biosynthetic genes in hypocotyls, in a manner dependent on PHYTOCHROME-INTERACTING FACTORS (PIFs)".

Reviewer #2 (Remarks to the Author):

The authors have basically addressed all of my concerns and comments, and I have no further questions.

Reviewer #3 (Remarks to the Author):

I appreciate that the authors now include a targeted analysis of sterols and sphingolipids which shows there are no changes in the composition in response to the two light treatments (Figure S6). However, these data are relative and do not indicate whether there are more of these lipids as proposed based on the transcriptomics data and as suggested to allow for plasma membrane expansion. These new data also do not make the original data unseen in my mind which did not indicate changes in relative sphingolipid content, a major plasma membrane lipid. I agree that the data as shown in Figure 6 indicate an increase in phosphoglycerolipids which are likely in the plasma membrane given the small number of chloroplasts in hypocotyls. Absent of absolute quantification, I am still not convinced that in the overall picture of carbon partitioning in a hypocotyl sterol biosynthesis is a significant carbon sink, compared to cell walls or other carbon containing structures. Other interpretations of the role of sterols in this process are not ruled out by the current analysis.

The explanation for GGL in Figure legend 6 is misspelled.

Response to the reviewers' comments (NCOMMS-21-40466A)

Our answers are in **bold**

We would like to take this opportunity to thank the reviewers for their constructive comments.

Reviewer #1 (Remarks to the Author):

The revised manuscript by Ince et al. is significantly improved and they sufficiently addressed all of my concerns. However, I am with reviewer 3 in that the lipid analysis data do not fully support their conclusion. I think that the authors should tone down and rephrase their description and conclusions about lipid analysis. For example, instead of saying "LRFR promotes anabolism including biosynthesis of plasma-membrane sterols downstream of PHYTOCHROME-INTERACTING FACTORS (PIFs) acting in hypocotyls (lines 33-35)", they could describe "LRFR specifically induced expression of sterol biosynthetic genes in hypocotyls, in a manner dependent on PHYTOCHROME-INTERACTING FACTORS (PIFs)".

We thank you for the suggestion to discuss our data more carefully. To address this comment we made text adjustments to the abstract, results and discussion.

Reviewer #2 (Remarks to the Author):

The authors have basically addressed all of my concerns and comments, and I have no further questions.

We thank the reviewer for evaluation our work

Reviewer #3 (Remarks to the Author):

I appreciate that the authors now include a targeted analysis of sterols and sphingolipids which shows there are no changes in the composition in response to the two light treatments (Figure S6). However, these data are relative and do not indicate whether there are more of these lipids as proposed based on the transcriptomics data and as suggested to allow for plasma membrane expansion. These new data also do not make the original data unseen in my mind which did not indicate changes in relative sphingolipid content, a major plasma membrane lipid. I agree that the data as shown in Figure 6 indicate an increase in phosphoglycerolipids which are likely in the plasma membrane given the small number of chloroplasts in hypocotyls. Absent of absolute quantification, I am still not convinced that in the overall picture of carbon partitioning in a hypocotyl sterol biosynthesis is a significant carbon sink, compared to cell walls or other carbon containing structures. Other interpretations of the role of sterols in this process are not ruled out by the current analysis.

We thank reviewers 1 and 3 for their suggestion to moderate our claims. In response, to this suggestion we made modifications to the abstract, results and discussion. We do not intend to say that sterol biosynthesis is a particularly large carbon sink in hypocotyls of LRFR-treated seedlings. As we showed previously in de Wit et al., 2018 a significant fraction of carbon allocated to hypocotyls in LRFR-treated seedlings is in the lipid fraction. Taken together with our data from untargeted lipidomic analysis and cell biology experiments (Figure 6), this is consistent with the production of more PM lipids which includes sterols. We envisage sterols as an example of this increased demand for PM lipids. However, we do not claim that our work shows that increased

sterol biosynthesis required for plasma membrane growth is required for hypocotyl elongation. This is one hypothesis based on our data but in the discussion we also discuss other possible roles of sterols in hypocotyl growth.